# Ultrahigh-Pressure Metamorphism and P-T Path of Xiaoxinzhuang Eclogites from the Southern Sulu Orogenic Belt, Eastern China, Based on Phase Equilibria Modelling

Haiqi Yuan [1,2], Jian Wang [1,2,]* and Keiko Hattori [3,]*

1    College of Earth Sciences, Jilin University, Changchun 130061, China; yuanhq20@mails.jlu.edu.cn
2    Key Laboratory of Mineral Resources Evaluation in Northeast Asia, Ministry of Natural Resources of China, Changchun 130026, China
3    Department of Earth and Environmental Sciences, University of Ottawa, Ottawa, ON K1N 6N5, Canada
*    Correspondence: wangjian304@jlu.edu.cn (J.W.); khattori@uottawa.ca (K.H.)

**Abstract:** Three types of eclogites were identified in the Xiaoxinzhuang area in the northern Sulu ultrahigh pressure (UHP) terrene based on their petrographic, compositional characteristics and locations. They are composed of garnet, omphacite, amphibole, epidote, phengite, quartz/coesite, rutile, apatite, ilmenite and kyanite. Garnet in eclogite exhibits weak compositional zoning, which shows an increase in $X_{gr}$ and a decrease in $X_{py}$ from core to mantle, and a decrease in $X_{gr}$ and a slight increase in $X_{py}$ from mantle to rim. Phengite inclusions in garnet show higher Si, up to 3.424 p.f.u., than those in the matrix. Pseudosections calculated using THERMOCALC in the NCKFMASHTO system for three representative samples record three stages of metamorphism: (I) prograde stage, (II) post- $P_{max}$ decompression and heating to the $T_{max}$ stage and (III) retrograde stage. Stage-I was recorded in garnet cores with mineral assemblage of garnet + omphacite ± amphibole ± lawsonite + phengite + quartz + rutile, and the P-T condition is constrained at 23.5–26.4 kbar and 623–655 °C. The $P_{max}$, 41.5 kbar at 801 °C, is revealed from garnet enclosed by coarse-grained garnet with the mineral assemblage of garnet + omphacite + phengite + coesite + rutile. Stage-II produced garnet rim with mineral assemblage of garnet + omphacite + amphibole + quartz + rutile + metabasite melt, which constrained the P-T conditions of 21.4–23.0 kbar and 869–924 °C. Stage-III, recorded by unzoned garnet grain with the mineral assemblage of garnet + omphacite + amphibole + ilmenite + rutile + metabasite melt, constrained P-T conditions of 13.5–16.4 kbar and 813–852 °C. The data suggest that the rocks in the Xiaoxinzhuang area were subducted to a depth of over 135 km and underwent an UHP metamorphism. The P-T-t path revealed by the Xiaoxinzhuang eclogites is different from those in other areas of the Sulu UHP terrane, suggesting that they represent different rock slices during the subduction and exhumations.

**Keywords:** continental subduction; continental collision; UHP eclogite; P-T pseudosection

## 1. Introduction

The Sulu-Dabie UHP metamorphic belt in eastern China is the longest UHP terrane in the world, which was formed by the subduction of the Yangtze craton beneath the North China craton (NCC) during the middle-late Triassic period [1–4]. Previous studies show varying P-T conditions for the Sulu UHP rocks. Zhang et al. [5] estimated the peak P-T conditions of 30–39 kbar and 740–830 °C using the garnet–omphacite–coesite–phengite geothermobarometry of Ravna and Terry [6] and the $Fe^{2+}$-Mg exchange thermometry between garnet and clinopyroxene. Wang et al. [7] obtained the peak P-T conditions of $699 \pm 30$ °C at 45 kbar using Zr-in-rutile thermometry (with an uncertainty of $\pm 30$ °C). Ye et al. [8] considered that the subducted Yangtze craton reached > 200 km deep with peak P-T conditions up to 70 kbar and 1000 °C based on the exsolution of clinopyroxene

from garnet. The discrepancy of the peak P-T conditions may reflect differential subduction of slices in the Sulu orogenic belt, and different thermobarometries were used [9,10]. Previous studies on Sulu eclogites were carried out in Rongcheng [3], Yangkou Bay [4,8,11] and Donghai [3,12,13], located in the northern and southern parts of the Sulu UHP terrane, respectively. The P-T-t paths of Sulu eclogites are clockwise and characterized by decompression coupled with cooling, such as Yangkou and Donghai (CCSD) eclogites, or by decompression coupled with heating (Rongcheng) [3,4,11–13]. In addition, the peak pressure in the north is lower than that in the south. The eclogites in the Xiaoxinzhuang area in the southern part of the Sulu UHP terrane have not been examined. The study area is close to the Wulian—Qingdao—Yantai Fault (WQYF) that is considered as the northwestern boundary of the Sulu HP-UHP metamorphic belt. To evaluate spatial variations of metamorphic conditions in the whole Sulu metamorphic belt, the content of this study is an important supplement. This paper presents the petrography and mineral chemistry of eclogites from the Xiaoxinzhuang area and evaluates its P-T metamorphic evolution using THERMOCALC 3.45 by Powell et al. [14] and as used in Wei et al. [15].

## 2. Geological Setting

The Paleo-Tethys oceanic lithosphere subducted northward beneath the NCC, which was followed by the collision of Yangtze craton with NCC during the middle-late Triassic (~220 Ma), forming the longest HP-UHP metamorphic belt (Sulu–Dabie–Qinling HP-UHP metamorphic belt; over 2000 km in length) in the world [16–18]. The NNE-striking Tan-Lu fault displaced the eastern part of huge HP-UHP metamorphic belt for more than 500 km, forming the NE-SW trending Sulu HP-UHP belt [19,20].

The Sulu HP-UHP belt is further divided into a HP terrane in the south and an UHP terrane in the north (Figure 1a; [21–25]), which were unconformably overlain by the Cretaceous sedimentary strata and volcanoclastic rocks and were intruded by postorogenic Mesozoic granitic plutons [26,27]. The Sulu HP terrane predominantly consists of quartzites, quartz schists, blueschists, marbles and albite gneisses [4,21,26,28,29]. These rocks underwent peak metamorphism of 12–25 kbar and 500–600 °C [21,29]. The Sulu UHP terrane consisting mainly of granitic gneiss and metasedimentary rocks can be further subdivided into northern and southern parts (N-Sulu UHP terrane and S-Sulu UHP terrane) with the boundary of the Wulian–Qingdao–Yantai Fault (WQYF) fault. Similar to the metasedimentary rocks, which have mostly Neoproterozoic protolith ages (850~750 Ma) [20,30,31], granitic gneisses are also Neoprototerozic and have experienced UHP metamorphism together with the eclogites [16,32,33]. Numerous isolated bodies of coesite-bearing eclogite, amphibolite, garnet peridotite and metamorphosed mafic-ultramafic rocks occurred in these granitic gneisses [4,34–36]. Mafic-ultramafic rocks form boudins or bands with their long axes parallel or semi-parallel to the regional gneissic schistosity. Metasedimentary rocks including marble and jadeite quartzite are mostly lenticular or thin-layer in granitic gneiss [20]. Orthogneisses are monzonite and granite in composition (Figure 1b).

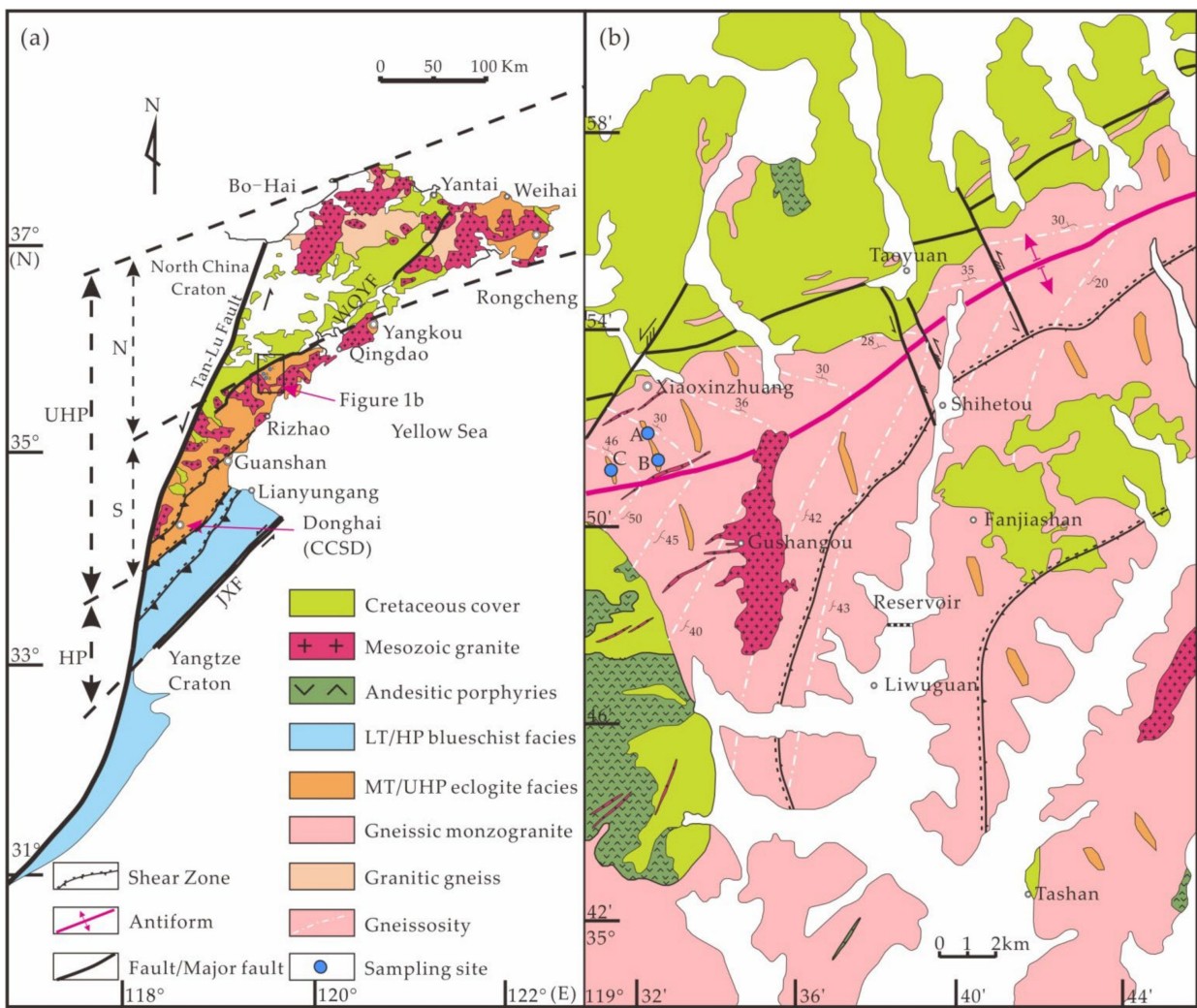

**Figure 1.** (**a**) Geological map of the Sulu orogenic belt, eastern China, showing the location of the Xiaoxinzhuang area (modified after [25]). S = southern part of Sulu UHP terrane; N = northern part of Sulu UHP terrane. (**b**) Simplified geological map of the Xiaoxinzhuang area with sampling sites on it (modified after [37,38]).

## 3. Sampling and Sample Petrography

The eclogite samples were collected in the Xiaoxinzhuang area, and sampling locations are shown in Figure 1b. Four samples (XX18-2, XX18-5, XX18-7 and XX18-9) in this study were collected 2–3 km south from the village of Xiaoxinzhuang. Two samples (XX18-8 and XX18-12) were collected 4–5 km southwest of Xiaoxinzhuang. Eclogite sporadically occurs as lenticular bodies with a length of 1–2 km in the gneissic monzogranite and extending along the northwest direction. Minerals of the Xiaoxinzhuang eclogites mainly include garnet, omphacite, amphibole, phengite and quartz/coesite. These eclogites underwent varying degrees of retrograde metamorphism. From the core to the mantle and to the rim of eclogite bodies, the retrograde metamorphism shows an increasing trend (Figure 2a). Outer margins (<30 cm) of eclogites are hydrated to form garnet amphibolites. The Xiaoxinzhuang eclogites are granoblastic and can be subdivided into three groups according to the sample locations and mineralogy.

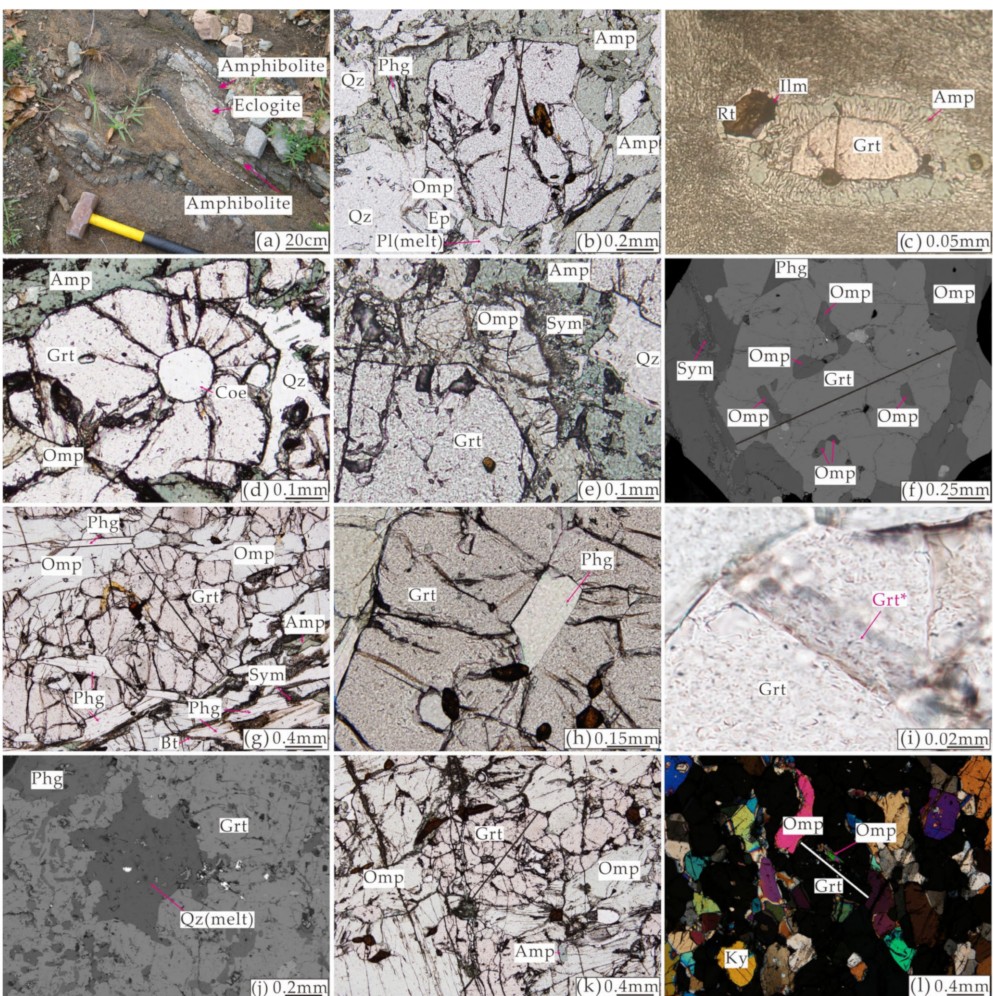

**Figure 2.** Photomicrographs showing mineralogy and textures for the Xiaoxinzhuang eclogites.
(**a**) Retrograde amphibolitization of a large eclogitic body. (**b**) Garnet occurs as porphyroblast
in the matrix of phengite, epidote, amphibole and plagioclase (Sample XX18-2). Note: A rutile
grain occurs as inclusion in garnet porphyroblast; Pl (melt) showing a wedge-shaped plagioclase
crystallized from a partial melt. (**c**) Coronal amphibole occurs as symplectite surrounding garnet
grain and rutile breakdown to form ilmenite (Sample XX18-2). (**d**) Coesite occurs as inclusion in host
mineral (garnet), which shows clear radial texture (Sample XX18-2). (**e**) Green amphibole formed
from omphacite during the retrograde metamorphism (Sample XX18-2). (**f**) Backscattered electron
image of garnet in Sample XX18-5. (**g**) Garnet occurs as porphyroblast in the matrix of omphacite,
phengite and amphibole (Sample XX18-5). Note: phengites are partially replaced by biotite and
symplectite due to retrograde metamorphism (**h**) Phengite occurs as inclusion in garnet (Sample
XX18-5). (**i**) Enlarged area of garnet porphyroblast with small tabular garnet (labelled with Grt*)
enclosed (sample XX18-5). (**j**): Star-shaped Qz (melt) formed from partial melt during prograde
metamorphism. (**k**) Garnet occurs as porphyroblast in the matrix of amphibole and omphacite
(Sample XX18-7). (**l**) Subhedral garnet grain with inclusions of omphacite under crossed polarizers
(sample XX18-12). The black or white lines across garnet grains in (**b,f,g,k,l**) show the locations of
the zoning profile. Mineral abbreviations: Amp = amphibole/pargasite; Ep = epidote; Grt = garnet;
Ky = kyanite; Omp = omphacite; Phg = phengite; Pl = plagioclase; Qz = quartz; Coe = coesite;
Rt = rutile; Sym = symplectite after omphacite or phengite.

Eclogite-I (samples XX18-2 and XX18-5) collected at sampling site A in Figure 1b
contains garnet (0.2–2.0 mm; 40–50 vol.%), omphacite (0.2–1.0 mm; 25–40 vol.%), amphi-
bole (0.5–1.0 mm; 5–15 vol.%), phengite (0.05–0.1 mm; 5–10 vol.%), rutile (0.1–0.2 mm;
2–3 vol.%), quartz (0.1–0.5 mm; 1–2 vol.%) and epidote (1.0–2.0 mm; <1 vol.%) (Figure 2b–j).

Garnet is subhedral to anhedral. It commonly contains inclusions of omphacite, phengite, quartz/coesite and rutile (Figure 2b,d,f,h,l). Tabular dark colored tabular garnet (Grt*, Figure 2i) occurs as small inclusions in coarse-grained garnet. Omphacite is subhedral to anhedral and is surrounded by the symplectitic mixture of amphibole + fine-grained albite + Fe-oxide (Figure 2e). Pseudomorphic coesite occurs as inclusions in host garnets, which show clear radial texture (Figure 2d). Amphibole formed during retrograde metamorphism. It occurs in two forms: matrix and corona. Amphiboles in the matrix are anhedral and flaky in texture and occur as veinlets in between garnet and omphacite grains (Figure 2b,c,e). The textures of green amphibole formed from omphacite due to decompression are common (Figure 2b,e). Coronal amphibole occurs along microfractures within garnet grains and coating of garnet, which formed from the breakdown of garnet (Figure 2c). Phengite flakes appear as euhedral-subhedral small grains in the matrix and distribute along the foliation together with rutile (Figure 2g). Phengite and rutile breakdown to form biotite and ilmenite, respectively, during the retrograde metamorphism (Figure 2c,g). Star-like quartz and wedge-shaped plagioclase can be found between the grains of other minerals, indicating that they formed from patches of melt during prograde metamorphism (Figure 2b,j).

Eclogite-II (samples XX18-7 and XX18-9), collected at site B of Figure 1b, contains garnet (0.5–2.0 mm; 55–60 vol.%), omphacite (0.5–1.5 mm; 35–40 vol.%), amphibole (0.5–1.0 mm; 2–3 vol.%), rutile (0.1–0.3 mm; 1–2 vol.%) and quartz (0.1–0.5 mm; <1 vol.%) (Figure 2k). Samples of eclogite-II are free of phengite relative to eclogite-I.

Eclogite-III (samples XX18-8 and XX18-12), collected at site C of Figure 1b, contains garnet (0.2–1.5 mm; 45–50 vol.%), omphacite (0.2–1.0 mm; 35–45 vol.%), kyanite (0.2–0.5 mm; 10–15 vol.%), rutile (0.1–0.2 mm; <1 vol.%) and quartz (0.1–0.5 mm; <1 vol.%) (Figure 2l). Samples of eclogite-III are free of phengite and amphibole relative to eclogite-I and II.

Based on the mineralogy and textures of eclogites in the study area, three mineral parageneses can be identified: Paragenesis-1 is composed of garnet + omphacite + phengite in which ompacite and phengite are characterized as inclusions in garnet porphyroblast. Minerals in paragenesis-1 represent products of prograde metamorphism (Figure 2f,h,l). Paragenesis-2 is characterized by garnet-mantle + omphacite + amphibole + phengite + melt in which melt formed from the breakdown of omphacite and phengite (Figure 2b,g,j), representing the stage of initial exhumation. Paragenesis-3 is composed of garnet-rim + omphacite + amphibole + ilmenite in which ilmenite formed from the breakdown of rutile (Figure 2c), corresponding to the stage of retrograde metamorphism.

## 4. Analytical Methods

The bulk-rock chemical compositions were obtained at Yanduzhongshi Geological Analysis Laboratories Ltd., Beijing, China. Unweathered samples were crashed to centimeter sizes and fresh pieces were selected, washed with deionized water, dried and then pulverized using an agate dish to less than (0.5200 ± 0.0001 g) for geochemical analyses. Sample powders were mixed with $Li_2B_4O_7$ (1:8) to make glass disks at 1250 °C using a V8C automatic fusion machine produced by the Analymate Company in China. Major element abundance was determined in fused glass disks using a Rigaku RIX 2100 XRF. FeO content was determined by the $Fe^{2+}$ titration method. Loss on ignition was determined by placing 1 g of samples in a furnace at 1000 °C for several hours. Based on rock standards AGV-2, GRS-1 and GSR-3 from USGS and Chinese national rock standards, the analytical precision and accuracy for major elements are better than 1%. Bulk-rock composition data are given in Table 1.

The modal abundances of minerals are obtained by point counting (total 4000 points/sample) and are presented in Table 2. P-T conditions are constrained using isopleths of pyrope ($X_{py}$) and grossular ($X_{gr}$) contents in garnet and Si contents in phengite in pseudosections.

**Table 1.** Bulk-rock compositions (wt.%) of eclogites from the Xiaoxinzhuang area.

| Sample | $Al_2O_3$ | $SiO_2$ | CaO | $K_2O$ | $TFe_2O_3$ [a] | MgO | MnO | $Na_2O$ | $P_2O_5$ | $TiO_2$ | LOI [b] | Total |
|---|---|---|---|---|---|---|---|---|---|---|---|---|
| | | | | | XRF analyses (wt.%) | | | | | | | |
| XX18-2 | 15.4 | 47.1 | 8.98 | 0.34 | 15.0 | 6.91 | 0.22 | 2.46 | 0.28 | 2.49 | 0.32 | 99.5 |
| XX18-5 | 15.8 | 49.3 | 10.5 | 0.49 | 11.0 | 7.28 | 0.14 | 4.34 | 0.21 | 0.83 | 0.36 | 100.3 |
| XX18-7 | 15.1 | 44.1 | 12.0 | 0.05 | 18.2 | 7.05 | 0.15 | 2.06 | 0.06 | 1.54 | 0.23 | 100.5 |
| XX18-8 | 23.6 | 42.8 | 11.3 | 0.05 | 9.22 | 11.3 | 0.11 | 0.98 | 0.03 | 0.09 | 0.43 | 99.9 |
| XX18-9 | 15.1 | 45.5 | 12.1 | 0.07 | 17.1 | 7.35 | 0.10 | 1.62 | 0.08 | 0.98 | 0.21 | 100.2 |
| XX18-12 | 22.7 | 45.0 | 10.4 | 0.01 | 9.64 | 9.37 | 0.12 | 2.35 | 0.03 | 0.13 | 0.10 | 99.8 |

| Sample | | $H_2O$ | $Al_2O_3$ | $SiO_2$ | CaO | $K_2O$ | FeO | MgO | $Na_2O$ | $TiO_2$ | O |
|---|---|---|---|---|---|---|---|---|---|---|---|
| | | | Normalized on the basis of mole per cent (mol %) | | | | | | | | |
| XX18-2 | P/T- | excess | 9.92 | 51.4 | 10.1 | 0.24 | 12.3 | 11.2 | 2.61 | 2.05 | 0.01 |
| | MO | excess | 9.33 | 48.4 | 9.47 | 0.22 | 11.6 | 10.6 | 2.45 | 1.93 | 5.89 |
| | P-T | excess | 9.86 | 51.1 | 10.0 | 0.24 | 12.2 | 11.2 | 2.59 | 2.03 | 0.60 |
| XX18-5 | P-T | excess | 9.86 | 52.2 | 11.6 | 0.33 | 8.76 | 11.5 | 4.46 | 0.66 | 0.43 |
| XX18-7 | P-T | excess | 9.45 | 46.9 | 13.6 | 0.03 | 14.6 | 11.2 | 2.13 | 1.23 | 0.71 |
| XX18-5 [c] | P-T | excess | 10.8 | 51.5 | 13.3 | 0.33 | 7.87 | 11.8 | 3.60 | 0.67 | 0.43 |

Note: [a] LOI, loss on ignition. [b] $TFe_2O_3$ is reported as FeO. [c] Effective bulk-rock compositions.

**Table 2.** Main mineral components and modal proportion (vol.%) of the Xiaoxinzhuang eclogites.

| Sample | Garnet | Omphacite | Phengite | Amphibole | Kyanite | Epidote | Quartz/Coesite | Rutile | Plagioclase |
|---|---|---|---|---|---|---|---|---|---|
| XX18-2 | 50 | 25 | 5 | 15 | - | 1 | 2 | 2 | <1 |
| XX18-5 | 40 | 40 | 10 | 5 | - | - | 2 | 3 | - |
| XX18-7 | 60 | 35 | - | 3 | - | - | 1 | 1 | - |
| XX18-12 | 50 | 35 | - | - | 15 | - | <1 | <1 | - |

Note: "-" = absent.

Mineral compositions were determined using a JEOL JXA-8230 electron microprobe at the Key Laboratory of Mineral Resources Evaluation in Northeast Asia, Ministry of Natural Resources, Jilin University (Changchun, China), using operating conditions of 15 kV acceleration voltage, 10 nA beam current and 1 μm beam diameter (5 μm for phengite). PRZ correction procedure was used to obtain raw data, and 53 standard minerals from the SPI Company were used for standardization. The representative mineral analyses are presented in Table 3 and Supplementary Materials Tables S1–S3. The mineral compositions were calculated using the programme AX (Holland; http://www.esc.cam.ac.uk/astaff/holland/ax.html accessed on 25 October 2021).

**Table 3.** Selected microprobe analyses for sample XX18-2 from the Xiaoxinzhuang eclogites.

| Mineral | Grt | | | | | | | | | | Omp | | Phg | | | | Amp | | Ep | | Ab |
|---|---|---|---|---|---|---|---|---|---|---|---|---|---|---|---|---|---|---|---|---|---|
| | Rim | Mid | Mid | Mid | Core | Mid | Mid | Mid | Rim | Rim | Inc | Grain | Inc [a] | Inc [a] | Core | Mid | Grain | Grain | Grain | Grain | Grain |
| $SiO_2$ | 39.4 | 39.5 | 39.4 | 39.5 | 39.9 | 39.2 | 39.2 | 39.1 | 39.3 | 39.6 | 55.7 | 55.9 | 49.7 | 48.9 | 48.7 | 49.6 | 45.3 | 44.3 | 38.4 | 38.9 | 66.4 |
| $TiO_2$ | 0.00 | 0.00 | 0.00 | 0.05 | 0.01 | 0.07 | 0.00 | 0.00 | 0.02 | 0.00 | 0.09 | 0.11 | 0.77 | 0.75 | 0.78 | 0.78 | 0.60 | 0.73 | 0.15 | 0.06 | 0.00 |
| $Al_2O_3$ | 22.4 | 22.2 | 22.2 | 21.9 | 22.2 | 22.1 | 22.0 | 22.0 | 21.8 | 22.0 | 11.4 | 11.4 | 27.6 | 27.8 | 27.7 | 27.6 | 13.0 | 12.7 | 26.7 | 27.0 | 19.8 |
| $Cr_2O_3$ | 0.02 | 0.01 | 0.06 | 0.06 | 0.06 | 0.05 | 0.04 | 0.05 | 0.05 | 0.02 | 0.09 | 0.00 | 0.03 | 0.04 | 0.00 | 0.04 | 0.05 | 0.06 | 0.02 | 0.03 | 0.04 |
| FeO [b] | 21.8 | 22.1 | 22.1 | 21.9 | 21.7 | 21.7 | 21.8 | 21.4 | 21.3 | 21.9 | 4.53 | 4.18 | 1.41 | 1.67 | 2.21 | 1.27 | 11.0 | 11.0 | 0.08 | 0.25 | 0.00 |
| $Fe_2O_3$ [c] | 0.00 | 0.00 | 0.00 | 0.00 | 0.00 | 0.00 | 0.00 | 0.42 | 0.00 | 0.00 | 0.00 | 0.00 | 0.83 | 0.43 | 0.00 | 1.20 | 1.48 | 1.61 | 8.53 | 7.57 | 0.05 |
| MnO | 0.50 | 0.51 | 0.55 | 0.46 | 0.39 | 0.46 | 0.46 | 0.52 | 0.39 | 0.47 | 0.04 | 0.00 | 0.01 | 0.01 | 0.00 | 0.03 | 0.16 | 0.11 | 0.22 | 0.12 | 0.00 |
| MgO | 7.61 | 7.66 | 7.59 | 7.67 | 7.56 | 7.36 | 7.29 | 7.40 | 7.42 | 7.50 | 7.81 | 7.61 | 3.01 | 2.83 | 2.40 | 2.97 | 10.8 | 10.9 | 0.17 | 0.21 | 0.03 |
| CaO | 8.19 | 7.85 | 8.07 | 8.37 | 8.07 | 8.65 | 8.54 | 9.03 | 8.55 | 8.46 | 13.3 | 12.7 | 0.00 | 0.02 | 0.00 | 0.06 | 8.53 | 9.30 | 22.5 | 23.2 | 1.87 |
| $Na_2O$ | 0.02 | 0.04 | 0.06 | 0.04 | 0.00 | 0.02 | 0.03 | 0.02 | 0.19 | 0.06 | 6.50 | 6.58 | 0.90 | 1.06 | 0.91 | 0.84 | 3.57 | 2.79 | 0.08 | 0.00 | 10.1 |
| $K_2O$ | 0.02 | 0.02 | 0.02 | 0.00 | 0.00 | 0.00 | 0.00 | 0.00 | 0.05 | 0.01 | 0.00 | 0.00 | 8.13 | 7.96 | 7.99 | 8.12 | 0.49 | 0.53 | 0.01 | 0.01 | 0.10 |
| Totals | 100.0 | 100.0 | 100 | 100.0 | 99.9 | 99.5 | 99.3 | 99.9 | 99.0 | 100.0 | 99.5 | 98.4 | 92.3 | 91.4 | 90.7 | 92.4 | 94.9 | 93.8 | 96.8 | 96.6 | 98.4 |
| | | | | | | | | | | Cations [d] | | | | | | | | | | | |
| Si | 3.01 | 3.02 | 3.01 | 3.02 | 3.04 | 3.01 | 3.02 | 3.00 | 3.03 | 3.03 | 1.99 | 2.01 | 3.37 | 3.36 | 3.37 | 3.36 | 6.74 | 6.68 | 3.03 | 3.04 | 2.96 |
| Ti | 0.00 | 0.00 | 0.00 | 0.00 | 0.00 | 0.00 | 0.00 | 0.00 | 0.00 | 0.00 | 0.00 | 0.00 | 0.04 | 0.04 | 0.04 | 0.04 | 0.07 | 0.08 | 0.01 | 0.00 | 0.00 |
| Al | 2.01 | 2.00 | 2.00 | 1.97 | 1.99 | 2.00 | 2.00 | 1.98 | 1.98 | 1.98 | 0.48 | 0.48 | 2.21 | 2.25 | 2.26 | 2.21 | 2.29 | 2.26 | 2.48 | 2.50 | 1.04 |
| Cr | 0.00 | 0.00 | 0.00 | 0.00 | 0.00 | 0.00 | 0.00 | 0.00 | 0.00 | 0.00 | 0.00 | 0.00 | 0.00 | 0.00 | 0.00 | 0.00 | 0.01 | 0.01 | 0.00 | 0.00 | 0.00 |
| $Fe^{3+}$ | 0.00 | 0.00 | 0.00 | 0.00 | 0.00 | 0.00 | 0.00 | 0.02 | 0.00 | 0.00 | 0.00 | 0.00 | 0.04 | 0.02 | 0.00 | 0.06 | 0.17 | 0.18 | 0.51 | 0.45 | 0.00 |
| $Fe^{2+}$ | 1.39 | 1.41 | 1.41 | 1.40 | 1.38 | 1.39 | 1.40 | 1.37 | 1.37 | 1.40 | 0.14 | 0.13 | 0.08 | 0.10 | 0.13 | 0.07 | 1.37 | 1.39 | 0.01 | 0.02 | 0.00 |
| Mn | 0.03 | 0.03 | 0.04 | 0.03 | 0.03 | 0.03 | 0.03 | 0.03 | 0.03 | 0.03 | 0.00 | 0.00 | 0.00 | 0.00 | 0.00 | 0.00 | 0.02 | 0.01 | 0.01 | 0.01 | 0.00 |
| Mg | 0.87 | 0.87 | 0.87 | 0.87 | 0.86 | 0.84 | 0.84 | 0.84 | 0.85 | 0.85 | 0.42 | 0.41 | 0.30 | 0.29 | 0.25 | 0.30 | 2.40 | 2.44 | 0.02 | 0.02 | 0.00 |
| Ca | 0.67 | 0.64 | 0.66 | 0.69 | 0.66 | 0.71 | 0.70 | 0.74 | 0.71 | 0.69 | 0.51 | 0.49 | 0.00 | 0.00 | 0.00 | 0.00 | 1.36 | 1.50 | 1.90 | 1.94 | 0.09 |
| Na | 0.00 | 0.01 | 0.01 | 0.01 | 0.00 | 0.00 | 0.00 | 0.00 | 0.03 | 0.01 | 0.45 | 0.46 | 0.12 | 0.14 | 0.12 | 0.11 | 1.03 | 0.82 | 0.01 | 0.00 | 0.87 |
| K | 0.00 | 0.00 | 0.00 | 0.00 | 0.00 | 0.00 | 0.00 | 0.00 | 0.01 | 0.00 | 0.00 | 0.00 | 0.70 | 0.70 | 0.71 | 0.70 | 0.09 | 0.10 | 0.00 | 0.00 | 0.01 |
| X (phase) [e] | 0.29 | 0.29 | 0.29 | 0.29 | 0.29 | 0.28 | 0.28 | 0.28 | 0.29 | 0.29 | 0.25 | 0.24 | 0.79 | 0.75 | 0.66 | 0.81 | | | | | |
| Y (phase) [e] | 0.23 | 0.22 | 0.22 | 0.23 | 0.23 | 0.24 | 0.24 | 0.25 | 0.24 | 0.23 | 0.47 | 0.48 | | | | | | | 0.17 | 0.15 | |

Note: [a] Inc = inclusions in garnet; [b] Total FeO is reported as FeO; [c] $Fe_2O_3$ is calculated by program AX; [d] Numbers of cation in Grt, Omp, Phg, Amp, Ep and Ab are calculated based on oxygens of 12, 6, 11, 23, 12.5 and 8, respectively, with the program AX; [e] X (g) = $X_{py}$ = Mg/($Fe^{2+}$ + Mn + Mg + Ca), X (o) = $Fe^{2+}$/($Fe^{2+}$ + Mg) on the basis of $Fe^{3+}$ = Na-Al-Cr; X (phn) = Mg/($Fe^{2+}$ + Mg); Y (g) = $X_{gr}$ = Ca/($Fe^{2+}$ + Mn + Mg + Ca), Y (o) = j (o) = Na/(Na + Ca), Y (ep) = $Fe^{3+}$/($Fe^{3+}$ + $Al^{VI}$).

## 5. Bulk-Rock and Mineral Composition

### 5.1. Bulk-Rock Compositions

The bulk-rock compositions are listed in Table 1. They show $SiO_2$ ranging from 42.79 to 49.33 wt.%, $Al_2O_3$ from 15.08 to 23.62 wt.%, MgO from 6.91 to 11.28 wt.%, $X_{Mg}$ [=MgO/(MgO + FeO) in mole] = 0.47–0.74 and CaO from 8.98 to 12.05 wt.%.

Eclogite-I samples have higher $SiO_2$ (47.1–49.3 wt.%) and alkalis ($K_2O + Na_2O$ = 2.81–4.83 wt.%) than eclogite-II ($SiO_2$ = 44.1–45.5 wt.%, $K_2O + Na_2O$ = 1.69–2.11 wt.%) and eclogite-III ($SiO_2$ = 42.8–45.0 wt%, $K_2O + Na_2O$ = 1.03–2.36 wt.%). Eclogite-I plot in the field of basalt, while eclogite-II and -III plot in the field of picrobasalt in the diagram of $SiO_2$ vs. ($K_2O + Na_2O$) for volcanic rocks by Middlemost et al. ([39] Figure 3a). The relatively higher $SiO_2$ and alkalis in eclogite-I are consistent with the presence of variable modal abundance of phengite. In addition, eclogite-I have $X_{Mg}$ = 0.52–0.61, which are higher than eclogite-II ($X_{Mg}$ = 0.47–0.50) but lower than eclogite-III ($X_{Mg}$ = 0.69–0.74). Furthermore, eclogite-I show lower CaO of 8.98–10.5 wt% but higher MnO of 0.14–0.22 wt.%, $TiO_2$ of 0.83–2.49 wt% and $P_2O_5$ of 0.21–0.28 wt% than eclogite-II (CaO = 12.0–12.1 wt.%, MnO = 0.10–0.15 wt.%, $TiO_2$ = 0.98–1.54 wt.% and $P_2O_5$ = 0.06–0.08 wt.%) and eclogite-III (CaO = 10.4–11.3 wt.%, MnO = 0.11–0.12 wt.%, $TiO_2$ = 0.09–0.13 wt.% and $P_2O_5$ = 0.03 wt.%; Table 1). In addition, eclogite-I show higher $K_2O$ (0.34–0.49 wt.%) and $K_2O$ /$Na_2O$ ratios (=0.11–0.14) than eclogite-II with $K_2O$ of 0.05–0.007 wt.% and $K_2O/Na_2O$ of 0.02–0.04 and eclogite-III with $K_2O$ of 0.01–0.05 wt.% and $K_2O/Na_2O$ of 0.01–0.05, respectively. Eclogite-I plot in the fields of "calc-alkali series" while eclogite-II and -III plot in the transitional fields of "low-K tholeiitic series" and "calc-alkali series" in the $K_2O$ vs. $SiO_2$ diagram by Peccerillo et al. ([40] Figure 3b).

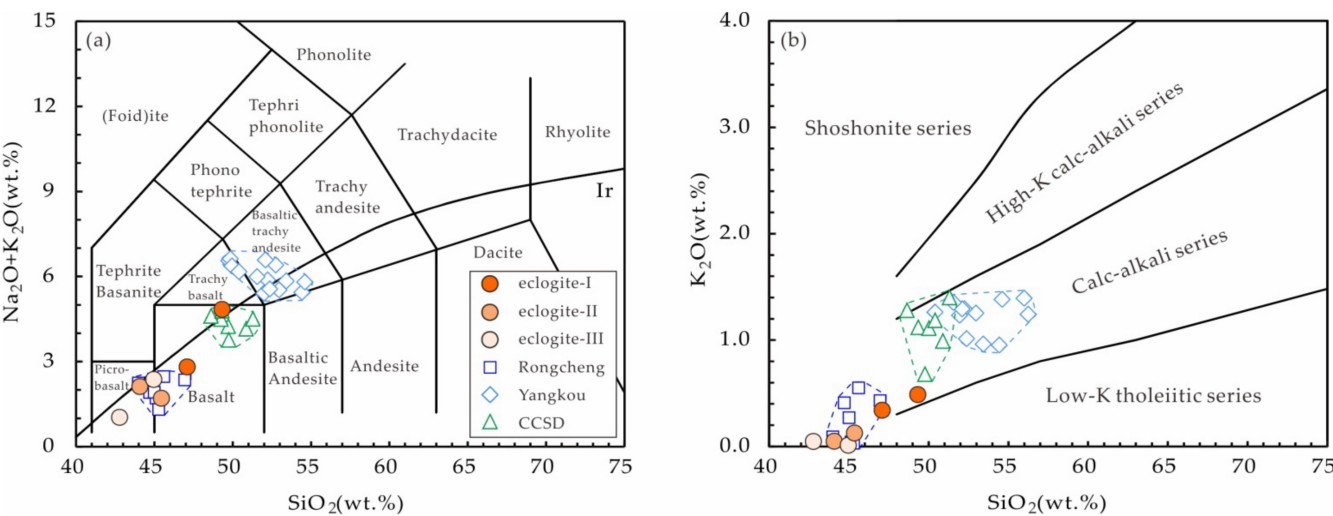

**Figure 3.** (**a**) Classification diagram of volcanic rocks based on total alkalis vs. silica by Le Bas et al. [41] and (**b**) $K_2O$ vs. $SiO_2$ (after [40]). Data of the Rongcheng eclogites are from [42]; data of Yangkou eclogite are from [33,43,44]; data of Chinese Continental Scientific Drilling (CCSD) eclogite are from [45].

Compositionally, eclogite-I samples are similar to the eclogites from the Chinese Continental Scientific Drilling (CCSD) [45], which are located in the southern part of S-Sulu UHP terrane, while eclogite-II and -III are similar to the eclogites from Rongcheng [42], which are located in the northern part of N-Sulu UHP terrane (Figure 3a,b). On the contrary, all the eclogites in the study area are distinctly different in composition from the Yangkou eclogites, which are located in the northern part of S-Sulu UHP terrane [33,43,44] and fall in the field of basaltic trachyandesite (Figure 3a).

### 5.2. Mineral Compositions

5.2.1. Garnet

Garnet exhibits zoning where, from core to mantle, $X_{gr}$ increases while $X_{py}$ decreases; from mantle to rim, $X_{gr}$ decreases while $X_{py}$ slightly increases (Figure 4). $X_{sps}$ shows a slight increase from core to mantle and then a slight decrease from mantle to rim (Figure 4a,c); meanwhile, $X_{alm}$ shows a slight decrease from core to mantle and then increases from mantle to rim (Figure 4). Tabular garnet (Grt*) shows higher contents of pyrope and grossularite and lower contents of almandine and spessartite than its host garnet porphyroblast (Grt) in sample XX18-5 (Figure 5a).

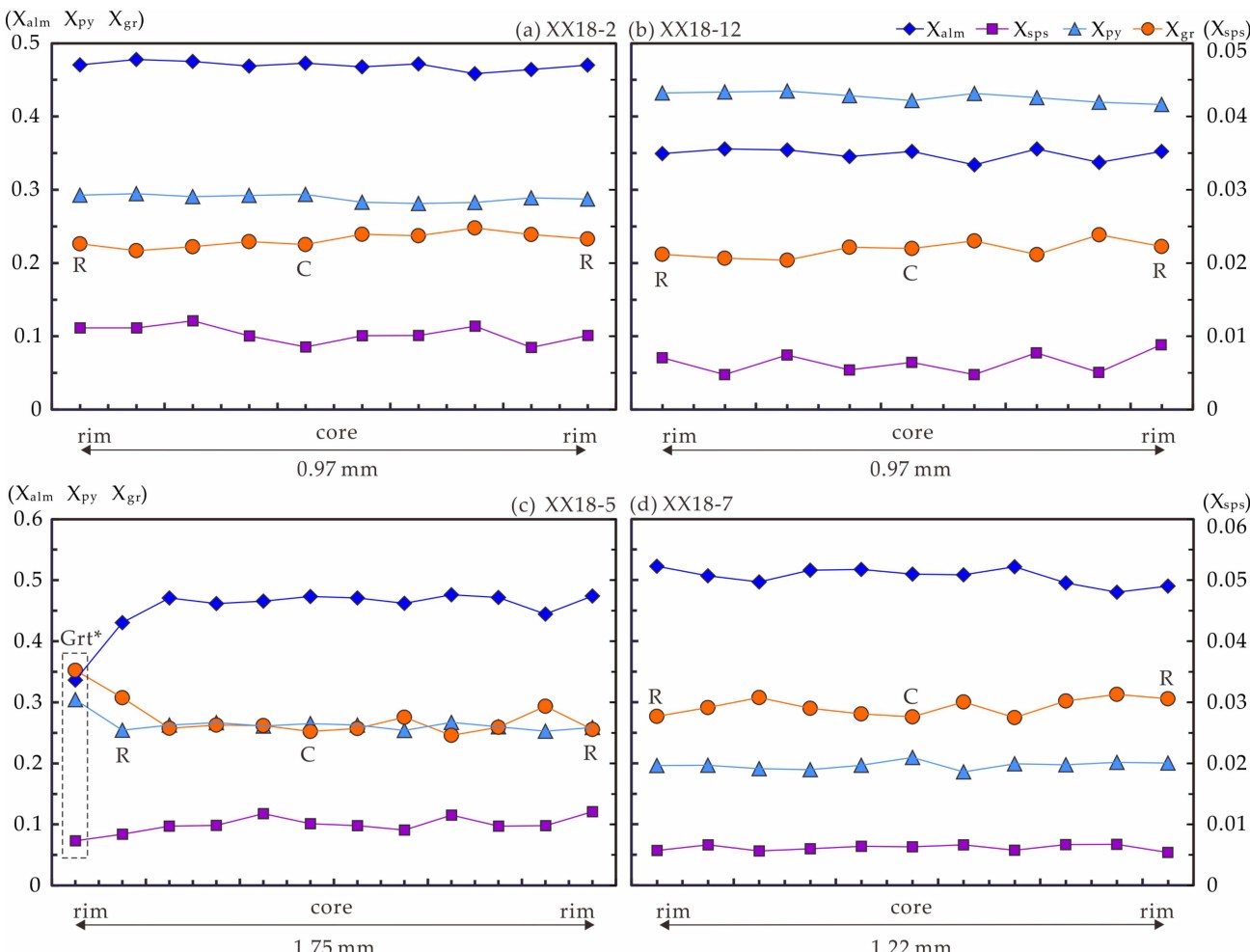

**Figure 4.** Zoning profiles of $X_{alm} = Fe^{2+}/(Fe^{2+} + Mn + Mg + Ca)$, $X_{sps} = Mn/(Fe^{2+} + Mn + Mg + Ca)$, $X_{py} = Mg/(Fe^{2+} + Mn + Mg + Ca)$ and $X_{gr} = Ca/(Fe^{2+} + Mn + Mg + Ca)$ across garnet grains from samples (**a**) XX18-2, (**b**) XX18-12, (**c**) XX18-5 and (**d**) XX18-7. Labels C, M, R correspond to those in Table 3 and Tables S1–S3. Left y-axis shows $X_{alm}$, $X_{py}$ and $X_{gr}$. Right y-axis shows $X_{sps}$. The locations of the compositional profile in garnet grains for Fig. a, b, c and d are shown as black or white lines in Figs. 2b, 2l, 2g and 2k, respectively.

Garnet shows a composition variation with $X_{py}$ ($Mg/(Fe^{2+} + Mg + Ca + Mn)$) ranging from 0.19 to 0.43, $X_{gr}$ ($Ca/(Fe^{2+} + Mg + Ca + Mn)$) from 0.20 to 0.35 and $X_{alm}$($Fe^{2+}/(Fe^{2+} + Mg + Ca + Mn)$) from 0.33 to 0.52. Garnet in sample XX18-2, XX18-5, XX18-7 and XX18-12 has compositions of $Alm_{46-48}Grs_{22-24}Prp_{29}Sps_1$, $Alm_{40-45}Grs_{29}Prp_{25-30}Sps_1$, $Alm_{50-51}Grs_{29-30}Prp_{20}Sps_1$ and $Alm_{34-35}Grs_{21-23}Prp_{42-43}Sps_1$, respectively (Figure 5b). The $X_{gr}$ values (=0.25–0.31) in XX18-5 and XX18-7 are higher than those ($X_{gr}$ = 0.20–0.25) in XX18-2 and XX18-12. The

low $X_{py}$ values, 0.19–0.20, are observed in XX18-7, whereas high $X_{py}$ values (0.42–0.43) are observed in XX18-12 (Tables 3 and S1–S3).

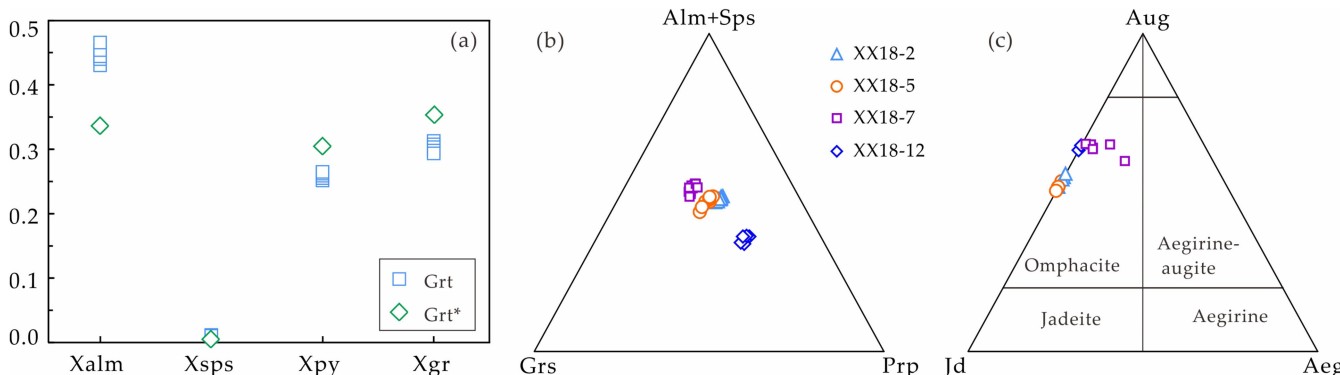

**Figure 5.** (**a**) The compositional map of tabular garnet (Grt*) and garnet porphyroblast (Grt) in sample XX18-5. (**b**) Ternary plots with apexes of almandine (Alm) + spessatine (Sps), grossular (Grs) and pyrope (Prp) showing the compositions of garnet in the Xiaoxinzhuang eclogites; (**c**) Ternary plots with apexes of jadeite (Jd), aegirine (Aeg) and augite (aug) showing compositions of omphacite in the Xiaoxinzhuang eclogites.

### 5.2.2. Omphacite

Omphacite compositions are shown in Figure 5c and Tables 3 and S1–S3. They are divided into two subgroups, with Omp-I corresponding to eclogite-I and Omp-II corresponding to eclogite-II and -III in the study area. Omp-I shows higher j (o) [=Na/(Na + Ca)] values of 0.47–0.53 and jadeite contents of 0.44–0.53 than that of Omp-II, which has j (o) values of 0.36–0.40 and jadeite contents of 0.25–0.36, respectively. The value of X (o) [=Fe$^{2+}$/(Fe$^{2+}$ + Mg)] of omphacite in the matrix is lower than omphacite as inclusions in Omp-I. On the contrary, the value of X (o) of omphacite in the matrix is higher than that of omphacite as inclusions in Omp-II. Except for the value of X (o), there is no significant compositional variation for the omphacite in the matrix or as inclusions.

### 5.2.3. Amphibole

Amphiboles are divided into two types. They are katophorite and pargasite, respectively, following the classification of Leake et al. ([46] Figure 6a). The amphibole in the matrix is katophorite (Si = 6.49–6.91, (Na)$_{M4}$ = 0.55–1.11); the Coronal amphibole is an Al-rich pargasite with Si of 5.65–5.67 p.f.u. (Table 3 and Tables S1 and S2).

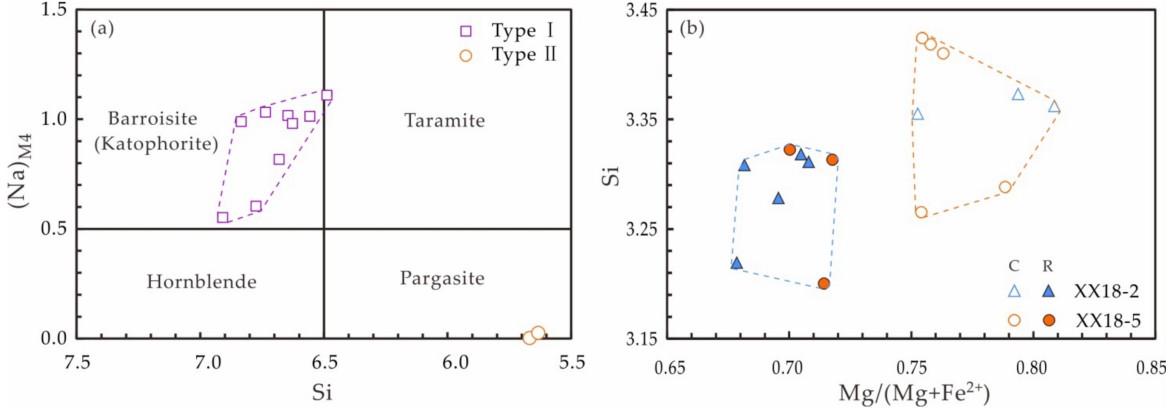

**Figure 6.** (**a**) Si versus (Na)$_{M4}$ diagram showing the classification of amphibole (modified after [46]); (**b**) Diagram of Mg/(Mg + Fe$^{2+}$) versus Si (11 oxygens) showing compositions of phengite in Xiaoxinzhuang eclogites. C = core; R = rim.

### 5.2.4. Phengite

Phengite has Si = 3.15–3.42, (Mg + Fe$^{2+}$) of 0.33–0.450 p.f.u. and Mg/(Mg + Fe$^{2+}$) = 0.66–0.81 (Tables 3 and S1). Phengite grains are zoned with Si decreasing from core to rim (Figure 6b), and phengite inclusions in garnet show higher Si and Mg + Fe$^{2+}$ than those in matrix.

### 5.2.5. Epidote

The pistacite (Ps) (=Fe$^{3+}$/(Fe$^{3+}$ + Al$^{VI}$)) contents for the epidote range from 0.12 to 0.17 (Tables 3 and S1).

## 6. Phase Equilibrium Modelling

Based on the mineral assemblages, mineral chemistry and bulk-rock compositions of eclogites, phase equilibria were modelled in the system NCKFMASHTO (Na$_2$O-CaO-K$_2$O-FeO-MgO-Al$_2$O$_3$-SiO$_2$-H$_2$O-TiO$_2$-O (Fe$_2$O$_3$)). Pseudosection calculations were performed using THERMOCALC 3.45 (updated 2016) [47] with the internally consistent thermodynamic dataset of the Holland and Powell [48] update (ds62). The initial bulk-rock compositions for constructing pseudosections are normalized by molar proportion, which are taken from XRF analysis of the weight percentage for each oxide (Table 1). P$_2$O$_5$ was disregarded because it mostly enters accessory apatite. CaO accounting for apatite was subtracted from the bulk-rock compositions. Mn content is neglected because it is very low (0.11–0.22 wt.%) and is mainly in spessartine in the Xiaoxinzhuang eclogites. In addition, minor associated Si and Al contents in spessartine were also taken out from the system. The value O (Fe$^{3+}$/(Fe$^{3+}$ + Fe$^{2+}$)) was included considering its effects on the stability of omphacite, amphibole and epidote. The O (Fe$^{3+}$/(Fe$^{3+}$ + Fe$^{2+}$)) content used in the modelling is adjusted using P–M$_O$ and T–M$_O$ diagrams to mitigate potential contamination or oxidation during sample preparation [49] and the effects of fluid that may cause changes in oxygen fugacity during exhumation [50,51]. Quartz/coesite and fluid phase are considered as in excess. Water may not be in excess, but here we do not consider it. The involved activity–composition relationships for the main solid solution phases are: metabasite melt [52], clinopyroxene [52], amphibole [52,53], garnet [54–57], muscovite [57], plagioclase [58], epidote [48], biotite [57], ilmenite [59] and talc [60]; and the pure end-member phases include lawsonite, kyanite, quartz, rutile, coesite and aqueous fluid (H$_2$O).

### 6.1. P-T Pseudosection for XX18-2 (Type I)

The stability of mineral assemblages with varying O contents from 0% Fe$^{3+}$ (MO = 0) to 99% Fe$^{3+}$ (MO = 1) over a range of 10 to 25 kbar at a fixed 700 °C is shown in Figure 7a, and that over a range of 650 to 850 °C at a fixed 20 kbar is presented in Figure 7b. The mineral assemblage observed in the sample Grt-Omp-Amp-Phg-Qz/Coe-Rt-LL is present in a quadrivariant assemblage field. Based on the petrographic observations, as the drop of pressure or the rise of temperature, the mineral assemblage should change to Grt-Omp-Amp-Qz/Coe-Rt-LL rather than Grt-Amp-Phg-Qz/Coe-Rt-LL or Grt-Omp-Amp-Phg-Rt-LL. Thus, any arbitrary value for MO < 0.3 can be used for further modeling. After many attempts, an A value of MO = 0.1 (O = 0.60 mol %) was selected to construct a P-T pseudosection. Similarly, the value of O = 0.43 mol % and 0.71 mol % were selected to construct a P-T pseudosection for samples XX18-5 and XX18-7, respectively.

The P-T pseudosection of sample XX18-2 is dominated by penta- and hexa-variant fields with a few quadri- and hepta-variant fields (Figure 8). In the assemblage of Grt-Omp-Amp-Phg (+Qz + Rt), isopleths for X$_{py}$ and X$_{gr}$ show similar negative slopes, and the rise of temperature is associated with an increase in X$_{py}$ and a decrease in X$_{gr}$. In addition, the Si content in phengite increases as pressure rises, which is considered as a good pressure indicator [15,61]. In the assemblages of Grt-Omp-Amp-LL (+Qz + Rt) and Grt-Omp-Amp-LL (+Rt), the isopleths of X$_{py}$ and X$_{gr}$ have moderate slopes, and, with the increasing pressure, X$_{py}$ increase while X$_{gr}$ decrease. In the assemblage of Grt-Omp-Amp-Ilm-LL (+Rt), the isopleths of X$_{gr}$ have flat slopes, and, with increasing pressure, X$_{gr}$

sharply decreases (a good indicator of pressure changes), while the $X_{py}$ (show moderate slopes) increases with temperature rise.

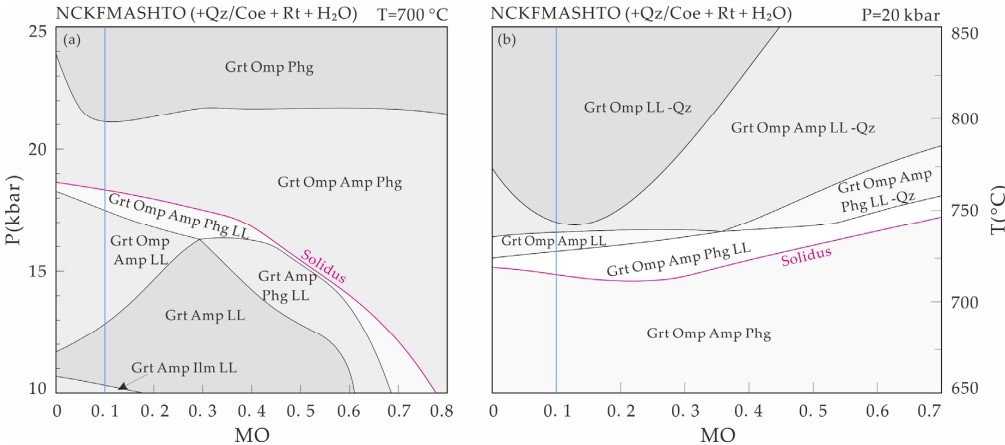

**Figure 7.** Diagrams of (**a**) P versus $M_O$ and (**b**) T versus $M_O$ for Sample XX18-2. Mineral abbreviations: Ilm = ilmenite; LL = mafic melt; other abbreviations are the same as in Figure 2. The compositions used for modelling are listed in Table 1. The higher variance fields are shaded with increasing intensity. "-Qz", denotes quartz-absent assemblages. The blue line represents the value of Mo we selected.

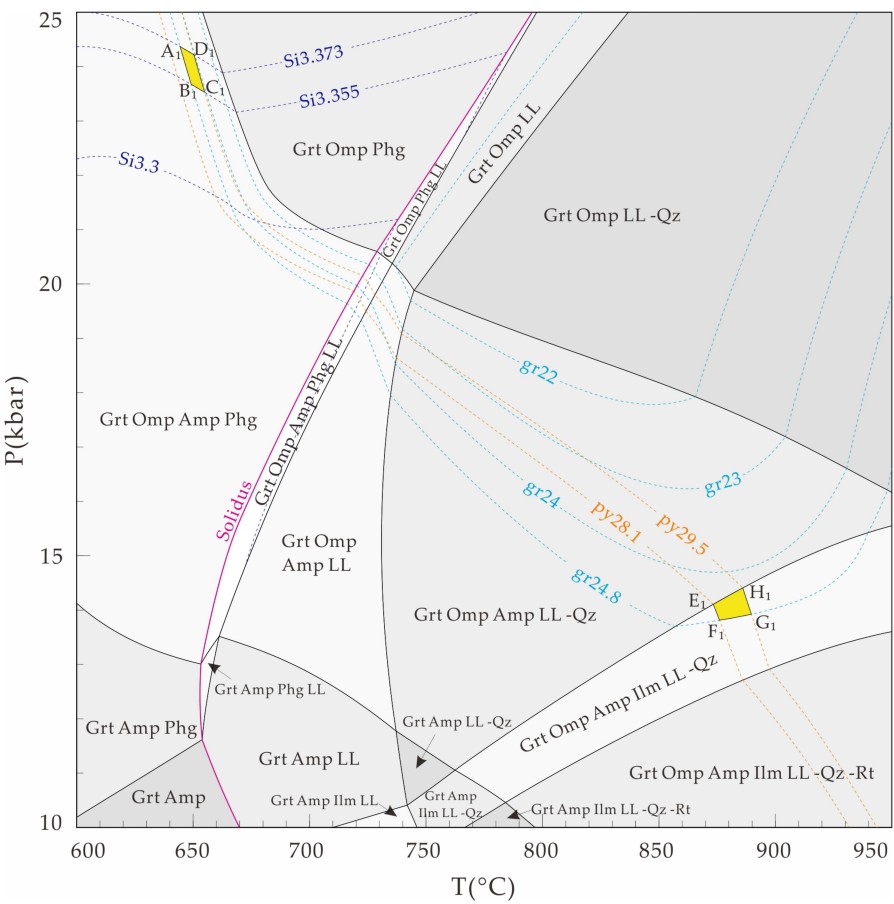

**Figure 8.** P-T pseudosection for sample XX18-2 calculated in the system NCKFMASHTO (+Qz/Coe + Rt + $H_2O$) using bulk-rock composition from Table 1. The pseudosection is contoured with isopleths of $X_{gr}$ (i.e., gr24.8) and $X_{py}$ (i.e., py28.1) in garnet and Si contents in phengite (i.e., s3.355). Mineral abbreviations: Bt = biotite; Lws = lawsonite; other abbreviations are the same as in Figure 2. "-Rt" denotes rutile-absent assemblages.

The minerals observed in sample XX18-2 include garnet, omphacite, amphibole, rutile, phengite and quartz, which constitute a penta-variant field in the model system with P-T conditions of 13–25 kbar and 600–720 °C (Figure 8). Based on mineral assemblages and the isopleths for $X_{gr}$ content (0.217–0.248) in garnet and Si content (3.355–3.373) in phengite, a field of $A_1$-$B_1$-$C_1$-$D_1$ is constrained at 23.5–24.4 kbar and 645–655 °C (Figure 8 and Table 3), which represents a prograde metamorphism before the peak stage. On the other hand, based on the contents of $X_{gr}$ (0.233–0.248) and $X_{py}$ (0.281–0.295) in garnet from mantle to rim (Table 3), a field of $E_1$-$F_1$-$G_1$-$H_1$ with mineral assemblage of garnet, omphacite, amphibole, metabasite melt, rutile and ilmenite is constrained at 13.8–14.4 kbar and 873–890 °C (Figure 8 and Table 4), which represents a decompression process (Figure 8). Under the P-T conditions of field $E_1$-$F_1$-$G_1$-$H_1$, the disappearance of phengite through reaction of Phg → Pl + Bt [12] was associated with the formation of amphibole and ilmenite through the breakdown of omphacite and rutile, respectively [62]. Furthermore, the presence of star-like quartz and wedge-shaped plagioclase indicates it has undergone partial melting during the prograde metamorphism or at the peak.

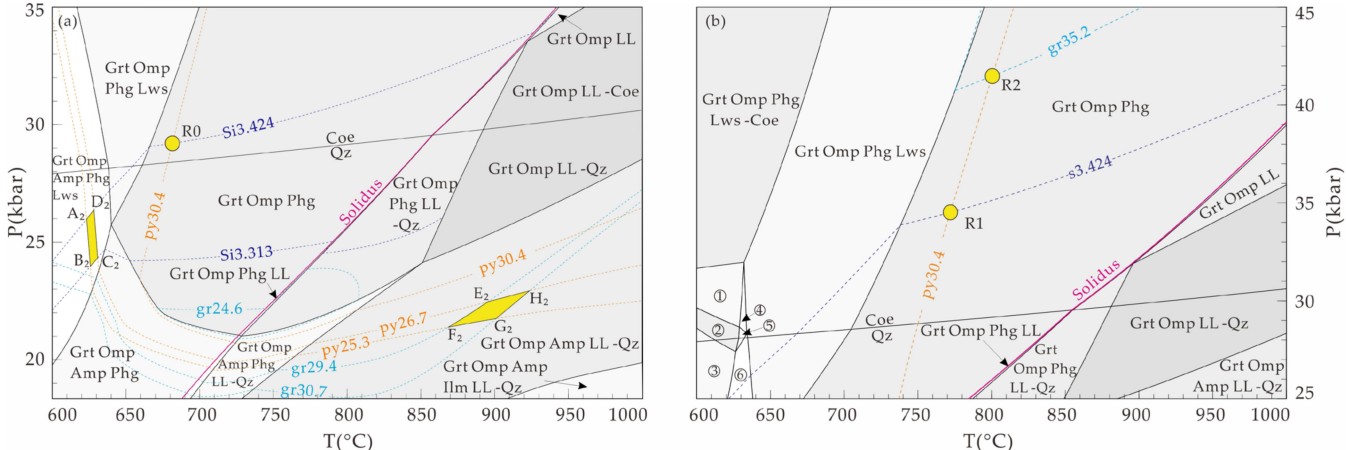

**Figure 9.** (**a**) P-T pseudosection for sample XX18-5 calculated in the system NCKFMASHTO (+Qz/Coe + Rt + $H_2O$) using bulk-rock composition in Table 1. (**b**) P-T pseudosection calculated in the system NCKFMASHTO (+Qz/Coe + Rt + $H_2O$) with an effective bulk composition generated according to mass-balance constraints from a type of garnet and other relevant minerals. Mineral abbreviations: Tlc = talc; other details are the same as in Figures 2 and 8. ① Grt Omp Phg Tlc Lws -Coe ② Grt Omp Amp Phg Tlc Lws -Coe ③ Grt Omp Amp Phg Lws -Coe ④ Grt Omp Phg Tlc Lws ⑤ Grt Omp Amp Phg Tlc Lws ⑥ Grt Omp Amp Phg Lws.

Table 4. Comparison of the calculated mineral compositions and modal proportions (on one-oxide basis) with those measured in the selected samples.

| | P (kbar) | T (°C) | py | gr | Si (mu) | Grt | Omp | Amp | Phg | Rt | Ilm | Coe/Qz | LL | Pl | Ep | Law |
|---|---|---|---|---|---|---|---|---|---|---|---|---|---|---|---|---|
| | | | | | | | Sample XX18-2 in Figure 8 | | | | | | | | | |
| $A_1$ | 24.4 | 645 | 0.29 | 0.25 | 3.37 | 0.51 | 0.32 | 0.04 | 0.04 | 0.02 | | 0.07 | | | | |
| $B_1$ | 23.7 | 649 | 0.29 | 0.25 | 3.36 | 0.51 | 0.32 | 0.04 | 0.04 | 0.02 | | 0.07 | | | | |
| $C_1$ | 23.5 | 655 | 0.29 | 0.24 | 3.36 | 0.51 | 0.33 | 0.03 | 0.04 | 0.02 | | 0.07 | | | | |
| $D_1$ | 24.2 | 650 | 0.29 | 0.24 | 3.37 | 0.51 | 0.33 | 0.03 | 0.04 | 0.02 | | 0.07 | | | | |
| $E_1$ | 14.1 | 873 | 0.28 | 0.24 | | 0.30 | 0.10 | 0.16 | | 0.02 | 0.00 | | 0.43 | | | |
| $F_1$ | 13.8 | 876 | 0.28 | 0.25 | | 0.28 | 0.09 | 0.18 | | 0.01 | 0.01 | | 0.43 | | | |
| $G_1$ | 13.9 | 890 | 0.29 | 0.25 | | 0.28 | 0.09 | 0.15 | | 0.01 | 0.01 | | 0.45 | | | |
| $H_1$ | 14.4 | 886 | 0.29 | 0.24 | | 0.31 | 0.10 | 0.13 | | 0.02 | 0.00 | | 0.44 | | | |
| Measured | | | 0.28–0.29 | 0.22–0.25 | 3.36–3.37 | 0.39 | 0.23 | 0.20 | 0.04 | 0.02 | | 0.08 | | 0.05 | <0.01 | |
| | | | | | | | Sample XX18-5 in Figure 9a | | | | | | | | | |
| $A_2$ | 26.0 | 623 | 0.25 | 0.26 | 3.42 | 0.30 | 0.51 | 0.08 | 0.05 | 0.01 | | <0.01 | | | | 0.05 |
| $B_2$ | 24.0 | 626 | 0.25 | 0.29 | 3.31 | 0.33 | 0.51 | 0.08 | 0.05 | 0.01 | | <0.01 | | | | 0.02 |
| $C_2$ | 24.3 | 631 | 0.27 | 0.28 | 3.31 | 0.34 | 0.53 | 0.05 | 0.05 | 0.01 | | 0.01 | | | | 0.01 |
| $D_2$ | 26.4 | 628 | 0.27 | 0.25 | 3.42 | 0.31 | 0.53 | 0.06 | 0.05 | 0.01 | | <0.01 | | | | 0.04 |
| $E_2$ | 22.5 | 896 | 0.27 | 0.29 | | 0.19 | 0.28 | 0.19 | | <0.01 | | | 0.34 | | | |
| $F_2$ | 21.4 | 869 | 0.25 | 0.29 | | 0.18 | 0.27 | 0.21 | | <0.01 | | | 0.34 | | | |
| $G_2$ | 21.8 | 901 | 0.25 | 0.31 | | 0.16 | 0.24 | 0.22 | | <0.01 | | | 0.38 | | | |
| $H_2$ | 23.0 | 924 | 0.27 | 0.31 | | 0.18 | 0.25 | 0.19 | | <0.01 | | | 0.37 | | | |
| R0 | 29.2 | 681 | 0.30 | 0.26 | 3.42 | 0.35 | 0.58 | | 0.05 | 0.01 | | 0.01 | | | | |
| Measured | | | 0.25–0.27 | 0.25–0.31 | 3.31–3.42 | 0.26 | 0.56 | 0.05 | 0.07 | 0.01 | | 0.02 | | 0.04 | 0.01 | |
| | | | | | | | Sample XX18-5 in Figure 9b | | | | | | | | | |
| R1 | 34.5 | 772 | 0.30 | 0.35 | 3.42 | 0.45 | 0.47 | | 0.05 | 0.01 | | 0.02 | | | | |
| R2 | 41.5 | 801 | 0.30 | 0.35 | 3.54 | 0.45 | 0.47 | | 0.05 | 0.01 | | 0.02 | | | | |
| Measured | | | 0.30 | 0.35 | 3.42 | 0.35 | 0.51 | 0.03 | 0.07 | 0.01 | | 0.02 | | | 0.01 | |
| | | | | | | | Sample XX18-7 in Figure 10 | | | | | | | | | |
| $A_3$ | 14.3 | 813 | 0.19 | 0.30 | | 0.37 | 0.21 | 0.26 | | <0.01 | 0.00 | | 0.15 | | | |
| $B_3$ | 13.5 | 819 | 0.19 | 0.31 | | 0.35 | 0.24 | 0.27 | | <0.01 | 0.01 | | 0.18 | | | |
| $C_3$ | 14.7 | 852 | 0.21 | 0.31 | | 0.38 | 0.20 | 0.20 | | <0.01 | 0.02 | | 0.20 | | | |
| $D_3$ | 16.4 | 847 | 0.21 | 0.29 | | 0.42 | 0.24 | 0.18 | | 0.01 | 0.00 | | 0.15 | | | |
| Measured | | | 0.19–0.21 | 0.27–0.31 | | 0.47 | 0.27 | 0.18 | 0.01 | 0.01 | | 0.01 | | 0.06 | | |

Note: $A_{1-3}$-$G_{1-2}$, R0, R1 and R2 correspond to labels in Figures 8–10. Other abbreviations are the same as those in Table 3.

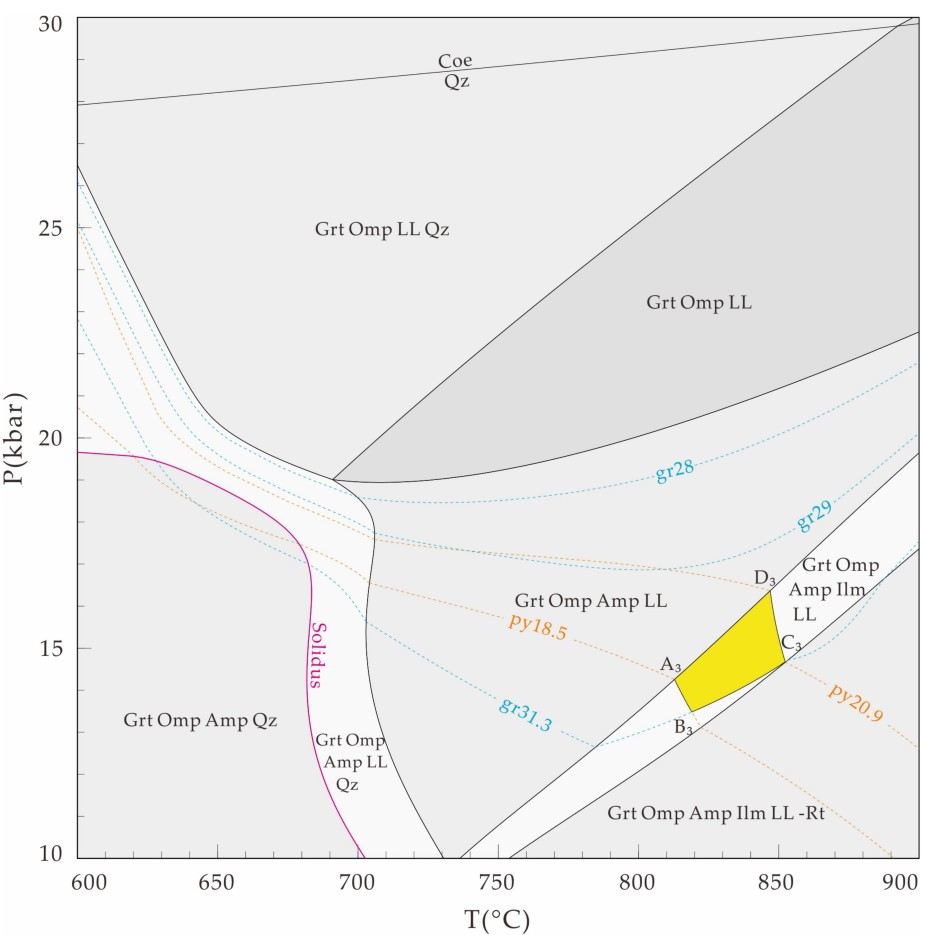

**Figure 10.** P-T pseudosection for sample XX18-7 calculated in the system NCKFMASHTO (+Rt + H$_2$O) using the bulk-rock composition in Table 1. Other details are the same as in Figures 2 and 8.

### 6.2. P-T Pseudosection for XX18-5 (Type I)

The observed minerals involving garnet, omphacite, amphibole and phengite (+quartz + rutile) are stable in the penta-variant field Grt-Omp-Amp-Phg (+Qz + Rt) with a pressure of 13.5–25.5 kbar and a temperature of 600–730 °C (Figure 9a). Furthermore, the isopleths of the measured X$_{py}$ and X$_{gr}$ in garnet in the assemblage Grt-Omp-Amp-Phg (+Qz + Rt) and Grt-Omp-Amp-Phg-LL (+Rt) show similar and flat negative slopes and increasing pressure results in the sharp increase in X$_{py}$ and the decrease in X$_{gr}$. In the assemblage Grt-Omp-Amp-Phg-Lws (+Qz/Coe + Rt), X$_{py}$ have steep slopes, and, with a rise in temperature, X$_{gr}$ increases sharply (a good indicator of temperature changes), while X$_{py}$ show flat slopes, and, with a rise in pressure, X$_{py}$ decrease. Si contents in phengite have moderate slopes and increase as pressure rises. In the assemblage Grt-Omp-Amp-LL (+Rt), X$_{py}$ and X$_{gr}$ have flat and moderate slopes, respectively, and, with a rise in pressure, the increasing X$_{py}$ is associated by decreasing X$_{gr}$.

A field of A$_2$-B$_2$-C$_2$-D$_2$ is constrained at a pressure of 24.0–26.4 kbar and at a temperature of 623–631 °C in the quadrivariant assemblage field Grt-Omp-Amp-Phg-Lws (+Rt + Qz) by the isopleths for X$_{py}$ (0.253–0.267) in garnet and Si content (3.313–3.424) in phengite (Table S1). Using the isopleths of the maximum Si content in phengite and the maximum X$_{py}$ in the rim of garnet is likely to define a peak P-T condition [61]. Accordingly, the isopleth of the maximum Si-isopleth (=3.424) in phengite with maximum X$_{py}$ (=0.304) in the garnet grain rim (Table S1) constrains a point (R0) of 29.2 kbar at 681 °C for the assemblage Grt-Omp-Phg (+Rt + Qz), reflecting a higher P and T condition during the pre-peak stage (Figure 9a).

A field of $E_2$-$F_2$-$G_2$-$H_2$ is constrained at a pressure of 21.4–23.0 kbar at T of 869–924 °C (Table 4), which have mineral assemblage of garnet, omphacite, amphibole, metabasite melt and rutile, constituting a penta-variant field in the model system, likely representing a process of decompression (Figure 9a). Compared with field $A_2$-$B_2$-$C_2$-$D_2$, phengite disappears, likely via the reaction Phg → Pl + Bt in the field $E_2$-$F_2$-$G_2$-$H_2$ [12]. However, ilmenite was not formed. The probable reason is that the pressure in this field is higher than the field in $E_1$-$F_1$-$G_1$-$H_1$ in Figure 8. The pressure of $E_2$-$F_2$-$G_2$-$H_2$ is higher than the transitional pressure from rutile to ilmenite. It indicates that the sample XX18-2 has a higher degree of retrograde metamorphism than sample XX18-5, and sample XX18-2 is closer to the surface during the decompression stage. The constrained field is consistent with the microscopic observation that abundant amphibole is present in sample XX18-2, which was likely formed through the reaction Omp + (Pl) → Amp + (Pl) symplectite and Grt → Pargasite + Fe-oxide [12].

It is worth noting that garnet inclusion (Grt*, Figure 2i) shows distinctly different compositions with higher $X_{py}$ and $X_{gr}$ than hosting garnet (Table S1), suggesting that Grt* may have retained different P-T conditions from garnet granule. In order to evaluate the P-T condition of this garnet, we reconstructed effective bulk rock compositions to calculate the P-T pseudosection and generated the effective bulk-rock compositions following mass balance constraints by integrating the mineral compositions and modal abundance information of the phases present, with the microprobe analyses. The calculated P-T pseudosection using the effective bulk composition is presented in Figure 9b.

In the assemblages Grt-Omp-Phg (+Rt + Coe), the isopleths of $X_{gr}$ in garnet and Si contents in phengite show moderately positive slopes and are chiefly controlled by pressure, while the isopleths of $X_{py}$ in garnet show steep positive slopes, which are mainly dominated by temperature. The isopleths of the measured maximum Si content (=3.424) in phengite and maximum $X_{py}$ (=0.304) in the tabular garnet (Table S1) define a P-T condition point (R1) of 34.5 kbar at 772 °C in the hexa-variant field Grt-Omp (+Phg + Rt + Coe). The maximum $X_{py}$ (=0.304) and maximum $X_{gr}$ (=0.352) also define a 41.5 kbar at 801 °C for the same field. The P-T condition constrained by the maximum $X_{py}$ and maximum $X_{gr}$ in garnet is much higher than that by the maximum Si content in phengite and maximum $X_{py}$ in garnet (Table S1), indicating that this tabular garnet records the highest P ($P_{max}$ in this paper) condition for the sample, implying that Si in phengite was likely modified during the retrogression stage.

*6.3. P-T Pseudosection for XX18-7 (Type II)*

The P-T pseudosection calculated for sample XX18-7 using the analyzed bulk-rock composition with rutile and aqueous fluid ($H_2O$) in excess with the P-T window of 10–30 kbar and 600–900 °C is presented in Figure 10. The observed minerals involving garnet, omphacite, amphibole, ilmenite (+rutile) is stable in the penta-variant field Grt-Omp-Amp-Ilm-LL (+Rt) with a pressure of 10–20 kbar and a temperature of 735–900 °C, which is the same as the observed assemblage under microscope. In most assemblages, the isopleths of $X_{gr}$ and $X_{py}$ have moderate slopes, and, with the rise in pressure, $X_{py}$ contents are increasing while $X_{gr}$ contents are decreasing. In the assemblage Grt-Omp-Amp-Ilm-LL (+Rt), the isopleths of $X_{py}$ have deep negative slopes, with $X_{py}$ increasing rapidly as temperature rises, while the isopleths of $X_{gr}$ show moderate slopes, with $X_{py}$ decreasing as pressure rises (Figure 10).

Based on the mineral assemblages and isopleths for the $X_{gr}$ (0.275–0.313) and $X_{py}$ (0.185–0.209) in garnet, a field is constrained at a pressure of 13.5–16.4 kbar at T of 813–852 °C (Table S2), which reflects a retrograde metamorphism during the decompression stage (Figure 10). Compared with the samples of XX18-2 and XX18-5, sample XX18-7 is devoid of phengite. Ilmenite was formed via the reaction Rt → Rt + Ilm intergrowth [62] at the decompression stage. The bulk-rock composition is used to model the metamorphic evolution, as the garnet is mostly unzoned in sample XX18-7, and the measured $X_{gr}$ and $X_{py}$ in garnet from core to rim do not show distinct difference (Table S2), indicating the

temperature and pressure condition is similar in garnet grain from core to rim. Meanwhile, this temperature and pressure condition is likely to reflect the P-T condition of the whole rock sample. It indicates that the P-T conditions tend to be stable, much closer to the decompression later stage.

## 7. Discussion

### 7.1. Metamorphic Evolution of the Xiaoxinzhuang Eclogites

Pseudosection could utilize the maximum information available from the studied samples, which allows the evolution of mineral assemblages to determine a P-T path [4,15,63–67]. Different assemblages can be preserved in the same rocks, and we can still apply an equilibrium view, provided we choose the equilibrium volumes wisely. In the metamorphic process (e.g., prograde and retrograde), the samples are considered to be a close system. Based on the petrographic observations, mineral analyses and phase equilibria modeling for the three representative samples, the metamorphic evolution can be divided into three stages:

(1)   Prograde Stage

The pre-peak condition is revealed from the core of garnet in samples XX18-2 and XX18-5. The P-T conditions at the pre-peak are estimated to be 23.5–26.4 kbar and 623–655 °C in the NCKFMASHTO system, and the mineral assemblage is Grt + Omp ± Amp ± Lws + Phg + Qz + Rt. The P-T values estimated with the pseudosection approach in this study are similar to the pressure (=20.4 kbars) estimated using the Grt-Cpx-Pl-Qz barometer [68] and the temperature (647 °C in sample XX18-2 and 643 °C in sample XX18-5 with the core composition of garnet) estimated using Cpx thermobarometers (Table S4; [69]).

The Si contents in phengite is regarded as a good pressure indicator [15,61]. As shown in the pseudosection (Figures 8 and 9a), the Si contents increase synchronously with the increase of pressure. However, the composition of phengite may have been variably reset during retrograde metamorphism; thus, only the maximum Si content in phengite is used for estimating the peak P-T conditions of eclogites [15]. The maximum Si content (=3.424) in phengite in sample XX18-5, together with the small tabular garnet inclusion, constrained the higher prograde metamorphic conditions 34.5 kbar and 772 °C (R1).

The $P_{max}$ conditions ($P_{max}$ = 41.5 kbar; T = 801 °C) are constrained by a small tabular garnet (Grt*) enclosed in a large garnet grain (Figure 2i) with a mineral assemblage of Grt + Omp + Phg + Coe + Rt (R2; Figure 11). Our data are overall consistent with the results of previous studies, which commonly considered the $P_{max}$ > 35–45 kbars at 700–850 °C [7,70–73]. For example, Zhang et al. [74] obtained the $P_{max}$ of 30–39 kbars at a temperature of 740–830 °C using the garnet–omphacite–coesite–phengite geothermobarometry of Ravna and Terry [6] which assumes $Fe^{3+}$ = Na–$Al^{VI}$ and uses the calibration of Krogh et al. [75]. However, the estimated highest subduction depths in this study, as well as other prior studies, are far below the depth (up to 7 GPa at 1000 °C) estimated based on the exsolution rods of clinopyroxene in garnet by Ye et al. [8].

(2)   Post-$P_{max}$ Decompression and Heating to the $T_{max}$ Stage

The $T_{max}$ condition is constrained by the garnet rim in sample XX18-5 to be 869–924 °C at a pressure of 21.4–23.0 kbar with the mineral assemblage of Grt + Omp + Amp + LL + Rt (Figure 11), higher than the temperature (826 °C) calculated using the Grt–Cpx thermometer (Tables 5 and S4; [76]).

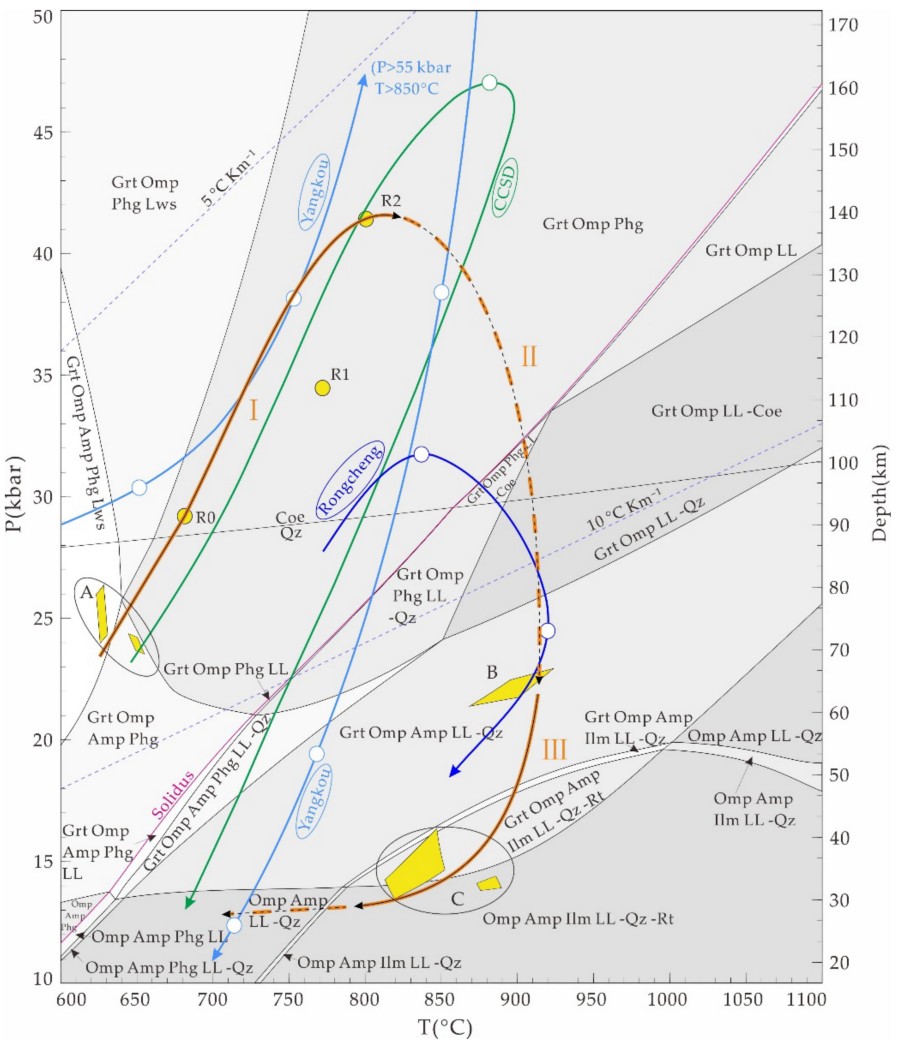

**Figure 11.** The P-T paths for the Xiaoxinzhuang eclogites and a comparison with those representing other areas of the Sulu UHP terrane. The base P-T pseudosection is based on the bulk composition of sample XX18-5 and calculated in the system NCKFMASHTO (+Qz/Coe + Rt + $H_2O$). The P-T-t path labelled with "Yangkou" is from [4] for the eclogites at Yangkou Bay, with "CCSD" is from [13] for the eclogites at the Chinese Continental Scientific Drilling (CCSD) deep borehole, and with "Rongcheng" is from [3] for the eclogites at Rongcheng. The yellow areas and circles (R0, R1 and R2) are the P-T ranges calculated by isopleth-thermobarometry of mineral compositions for Xiaoxinzhuang eclogites. I: Prograde Stage, II: Post-$P_{max}$ Decompression and Heating to the $T_{max}$ Stage, III: Retrograde Stage.

**Table 5.** A summary of P (kbar) –T (°C) estimates for the Xiaoxinzhuang eclogites.

| Sample | Geothermobarometers [a] | | Pseudosection [b] | | |
|---|---|---|---|---|---|
| | Prograde | Exhumation | Prograde | Peak | Exhumation |
| XX18-2 | 20.4 kbar/647 °C | 889 °C | 23.5–24.4 kbar/645–655 °C | | 13.8–14.4 kbar/873–890 °C |
| XX18-5 | 643 °C | 826 °C | 24.0–26.4 kbar/623–631 °C | 41.5 kbar/801 °C | 21.4–23.0 kbar/869–924 °C |
| XX18-7 | | 780 °C | | | 13.5–16.4 kbar/813–852 °C |

Note: [a] The conventional thermobarometers used here refer to the Grt-Cpx thermometer of [72], Cpx thermobarometers of [69], Grt-Cpx-Pl-Qz barometer of [68] and Grt-Cpx-Pl-Qz thermobarometers of [75]. [b] The P-T estimates for the prograde, peak and decompression stages were constrained using the isopleths of garnet and the phengite compositions in pseudosections (for details, see the text).

(3)     Retrograde Stage

The retrograde stage is inferred to be accompanied by considerable cooling, which is revealed from the garnet rim in sample XX18-2 and the unzoned garnet in sample XX18-7. The P-T conditions at the retrograde stage are estimated to be 13.5–16.4 kbar and 813–890 °C with the mineral assemblage of Grt + Omp + Amp + Ilm + LL + Rt (Figure 11). Decompression and cooling resulted in the formation of ilmenite from rutile in this stage (Figure 2c; [62]).

The estimated temperature from the pseudosection approach is consistent with the result (T = 889 °C) calculated by the Grt-Cpx-Pl-Qz thermobarometers [77] with the rim of garnet for sample XX18-2 and the result (T = 780 °C) by the Grt-Cpx thermometer [76] for sample XX18-7 (Table 5 and Table S4).

### 7.2. Comparison of P-T-t Path with Other Areas in the Sulu UHP Terrane

The P-T-t paths are clockwise in almost all UHP eclogites in the world [78]. The P-T-t paths of eclogites in the Sulu UHP terrane are also clockwise, as observed in the other parts of the Sulu UHP terrane (e.g., Rongcheng, Yangkou and CCSD), indicating that pressure increases followed by an increase in temperature. Compared with the eclogites from the other parts of the Sulu UHP terrane, the Xiaoxinzhuang eclogites are different in ranges of temperature and pressure and in the shape of the P-T-t path.

The Rongcheng eclogites from the northeastern part of the Sulu UHP terrane (Figure 1) show mineral assemblage of garnet, omphacite, amphibole, rutile, quartz/coesite, phengite and ilmenite [3,42]. Li et al. [3] calculated the metamorphic conditions based on pseudosection modelling using the Perple-X in the MnNCFMASHO system and obtained a maximum pressure ($P_{max}$) of 32 kbar at 840 °C and a peak temperature ($T_{max}$) of 910 °C at 24 kbar. The $P_{max}$ is low, but its $T_{max}$ and the style of P-T-t path are similar when compared with this study (Figure 11).

The Yangkou eclogites from the central Sulu UHP terrane have protolith of basaltic trachy andesite, which consists of garnet, omphacite, kyanite, phengite and quartz/coesite [4,79]. Previous workers obtained variable values of peak metamorphic conditions: 55 kbar/850 °C by Xia et al. [4], $T_{max}$ = 800–1000 °C by Wang et al. [7], 33 kbar/700 °C by Yang et al. [79] and 31.2–40.9 kbar/660–800 °C by Ye et al. [11]. Based on the phase equilibria modelling in the NC(K)FMASHTO system, Xia et al. [4] calculated the pseudosection for the Yangkou eclogites and obtained the P-T-t path. The $P_{max}$ of Yangkou eclogites is much higher than that of the Xiaoxinzhuang eclogites, but its $T_{max}$ is remarkably lower than the latter (Figure 11).

The eclogites from the Chinese Continental Scientific Drilling (CCSD) deep borehole in the southern part of the Sulu UHP terrane (Figure 1a) show a wide range of bulk-rock composition from andesitic to picritic basalt with a mineral assemblage of garnet, omphacite, kyanite, zoisite, phengite, rutile, amphibole, quartz/coesite, apatite and zircon [62]. The peak P-T conditions of CCSD eclogites are estimated to be 40–50 kbar/800–900 °C [13], and the corresponding P-T-t path is similar to that of the Yangkou eclogites (Figure 11).

Eclogites from different areas in the same HP-UHP belt show different P-T-t paths, which may reflect that their protolithes have undergone different dynamic processes during the subduction and exhumation. Similar cases are also reported in other well-known HP-UHP metamorphic belts (e.g., Longmu Co-Shuanghu belt in central Qiangtang and the Alps) [80,81].

### 7.3. Subduction and Exhumation of the Yangtze Continental Crust

Eclogite records important information of tectonic processes during the subduction and exhumation of continental crust, which generally forms in subduction zones or continental collision zones [82,83]. Similar to the P-T-t path, the subduction and exhumation processes in the study area are also divided into three stages:

(a) Rapid subduction stage.

The prograde mineral assemblages and mineral compositions preserved in samples XX18-5 and XX18-2 recorded the changes of P-T conditions (A(P/T = 23.5–26.4 kbar/623–655 °C) → R0(P/T = 29.2 kbar/681 °C) → R1(P/T = 34.5 kbar/772 °C) → R2(P/T = 41.5 kbar/801 °C) (Figure 11) and their corresponding depths (A (80–85 km) → R0 (95 km) → R1 (115 km) → R2 (135 km)) (Figures 11 and 12a) during the rapid subduction stage. The average dP/dT for this stage was high, 10–11 kbar/100 °C (=2.5–3.0 °C/km). The occurrence of coesite in sample XX18-2, together with the P-T values of $P_{max}$ (P = 41.5 kbar; T = 801 °C) and the high dP/dT rate for the prograde metamorphic process, suggest that UHP metamorphism happened in the Xiaoxinzhuang area. Amphibole and lawsonite did not appear in this stage because earlier formed amphibole would convert to omphacite (Table S5), and dehydration reactions would consume most of the lawsonite under higher pressure and temperature. Lawsonite can only be preserved in eclogite from cold subduction zones [15,78,84].

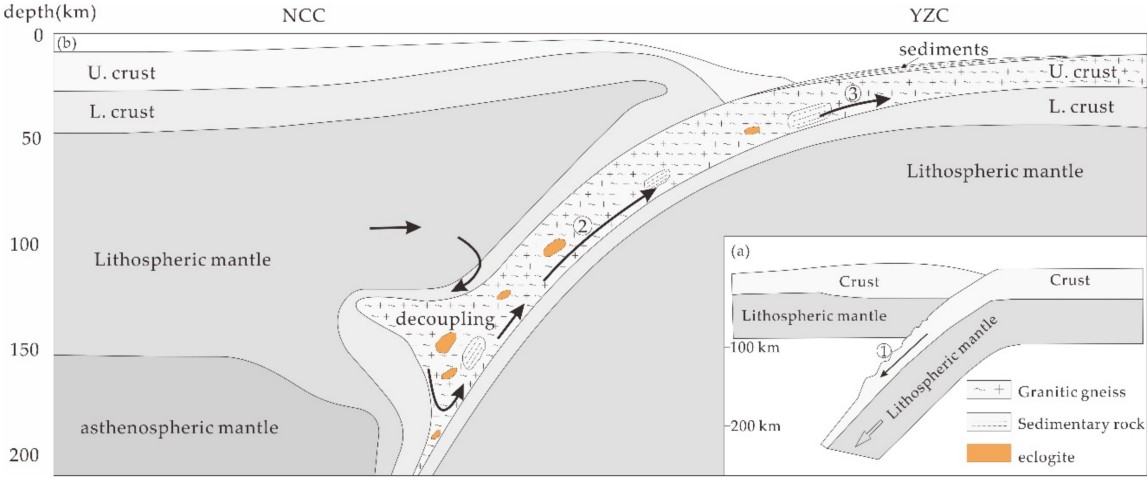

**Figure 12.** Schematic cross-sections showing the (**a**) subduction (after [85]) and (**b**) exhumation processes. The different depths of the studied samples are calculated based on the numerical models of Warren et al. [86]. U. crust = upper continental crust; L. crust = lower continental crust.

(b) Early exhumation stage.

The retrograde metamorphic mineral assemblages and mineral compositions preserved in sample XX18-5 recorded the changes of P-T conditions from R2 (P/T = 41.5 kbar/801 °C; Figures 11 and 12b) to B (P/T = 21.4–23.0 kbar/869–924 °C; Figure 11) and their corresponding depths from R2 (135 km) to B (70 km) (Figures 11 and 12b) during this stage. The average dP/dT for this stage was very high, up to 105–115 kbar/100 °C (=0.3 °C/km), which was considered as a near-isothermal decompression process. Fast exhumation resulted in the disappearance of coesite in this stage.

(c) Late exhumation stage.

The retrograde metamorphic mineral assemblages and mineral compositions preserved in sample XX18-5, -2 and -7 recorded the changes of the P-T conditions (from B (P/T = 21.4–23.0 kbar/869–924 °C) to C(P/T = 13.5–16.4 kbar/813–890 °C) (Figure 11) and their corresponding depths from B (70 km) to C (45–55 km)) (Figures 11 and 12b) during this exhumation stage. Compared with Stage ②, the geothermal gradients of this stage are higher, with an average dP/dT of 4–6 kbar/100 °C (=5.0–7.5 °C/km). This stage is characteristic of the disappearance of phengite and the appearance of lots of retrograde metamorphic minerals. Most groundmass amphiboles (green) formed from the breakdown of omphacite through the reaction of Omp + (Pl) → Amp + (Pl) symplectite (Figure 2e; [12]), and those corona amphiboles (pargasite in composition) formed from the breakdown of garnet through the reaction of Grt → Prg (+Fe-oxide) along the boundaries of garnet grains during the decompression process (Figure 2c). Similarly, ilmenite formed from rutile due to decompression [62].

## 8. Conclusions

(a) Three types of eclogite exposed in the Xiaoxinzhuang area in the Sulu UHP belt can be classified, which show three metamorphic stages: (I) prograde stage, (II) post-$P_{max}$ decompression and heating to the $T_{max}$ stage and (III) retrograde stage.

(b) The P-T conditions of prograde process were recorded in garnet core and phengite. Small tabular garnet (Grt*, Figure 2i) enclosed in the large garnet grain preserved $P_{max}$ condition, 41.5 kbar. The $T_{max}$, 869–924 °C, is revealed from the rim of garnet in sample XX18-5.

(c) The maximum subduction depth for the Xiaoxinzhuang eclogites is approximately 135 km.

**Supplementary Materials:** The following are available online at https://www.mdpi.com/article/10.3390/min12020216/s1, Table S1: Selected microprobe analyses for sample XX18-5 from the Xiaoxinzhuang eclogites, Table S2: Selected microprobe analyses for sample XX18-7 from the Xiaoxinzhuang eclogites, Table S3: Selected microprobe analyses for sample XX18-12 from the Xiaoxinzhuang eclogites. Table S4: The P (kbar) –T (°C) estimates using conventional thermobarometers. Table S5: Comparison of mineral assemblages in different metamorphic stages between microscopic and Pseudosection.

**Author Contributions:** Conceptualization, J.W. and K.H.; data curation, H.Y. and J.W.; laboratory analysis, H.Y. and J.W.; funding acquisition, J.W.; investigation, H.Y. and J.W.; methodology, J.W. and K.H.; resources, J.W. and K.H.; writing—original draft, H.Y.; writing—review & editing, J.W. and K.H. All authors have read and agreed to the published version of the manuscript.

**Funding:** This research was funded by the National Natural Science Foundation of China (NSFC), grant number 41472051, to J.W.

**Acknowledgments:** We are grateful to Wu Ding of Peking University for helping with the Thermocalc software and for the constructive discussions about thermodynamic models.

**Conflicts of Interest:** The authors declare no conflict of interest.

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
