# Peer review of "Ultrahigh-Pressure Metamorphism and P-T Path of Xiaoxinzhuang Eclogites from the Southern Sulu Orogenic Belt, Eastern China, Based on Phase Equilibria Modelling"

_minerals, doi:10.3390/min12020216_

Round 1

Reviewer 1 Report

This manuscript reports petrography and thermodynamically modelled P-T (and not P-T-t, as reported in the title) evolution of three types of UHP eclogites from a locality, never studied in detail, of the Sulu belt. The aim of the work is to increase the P-T data, from new localities, to help to evaluate spatial variations of metamorphic conditions in the whole Sulu metamorphic belt.

I support the idea to increase the P-T data in the Sulu belt but, although the manuscript is quite well organized and written, at present the manuscript could be good (after a deep revision of the data) for journals that have lower impact factors than Minerals. Otherwise, the authors have to effort to move from the very local perspective and try to discuss the implications on the whole UHP Sulu unit.

Regarding the manuscript, at present, the interpretations are questionable, because not always supported by the data, and can be interpreted in a different way. In my opinion, the data do not support the idea that the three eclogite-types represent the prograde, thermal peak and retrograde stages, respectively, a single P-T path (Fig. 12). If this is the case, I expect that eclogite-I show minor re-equilibration during the subsequent stages, or that eclogite-II and eclogite-III contain some relics of the previous stages. However, the authors also suggest that the samples could have been “undergone different dynamic processes during the subduction and exhumation” (discussion). This is a general statement, without information on specific processes so actually, I do not understand how the authors interpret the distinct P–T conditions reached by the samples. Otherwise, if the P-T data are correct, how can the authors rule out the possibility that they are actually observing the evidence of a tectonic mélange formed during exhumation?

Regarding the data, not all those plotted in figures are reported in the tables (e.g., the end-member proportions in Grt). Moreoveer, I saw some inconsistences between the data reported in tables and figures and those reported in text (see below). The most relevant one regards the lack of evidence of partial (or incipient) melting of the studied samples although the proposed thermodynamic modelling reveals that some metamorphic stages should be occurred at P-T conditions well above the solidus of the system.

Below, I report comments that, I hope, will be seen as constructive.

TITLE

  • The title needs to be changed. The manuscript only reports P-T data, without analytical constraints on the ages (i.e., “t”) of the metamorphic events.

INTRODUCTION

  • The introduction is too short. The authors could better describe the different kinds of P-T paths reconstructed for the Sulu belt (i.e. those characterized by decompression coupled with cooling or by decompression coupled with heating).
  • In Fig. 1a, the authors subdivide the Sulu belt in “northern”, “middle”, and “southern” parts. It is the first time that I see this subdivision for the Sulu belt and the authors have to explain why the boundaries among these parts have this slope. I would have expected horizontal boundaries, or boundaries parallel to the main tectonic structures.

SAMPLING AND SAMPLE PETROGRAPHY

  • I suggest to add few more sentences describing the outcrops and, if possible, to add one or two pictures of the outcrops.
  • Here the authors subdivide the samples in three eclogite-types on the basis of their mineralogy. However, this distinction, that is highlighted in the abstract and in the conclusion, is only locally reported in tables, figures or other part of the text. 
  • Figure 2. The presence of preserved coesite is not demonstrated by Fig. 2b and c. In my experience, coesite inclusions with radial cracks in partly retrogressed rocks, such as those reported in this study, are relics within retrograde quartz. The authors should substitute these two photomicrographs with others at higher magnification.
  • Similarly, the presence of a Grt* (line 118) is not demonstrated. The magnification in Fig. 2f is too low to recognise the “tabular inclusion” (strange shape of a garnet relic…), and in Fig. 2h (it would be “g”, I suppose) the Grt* is chemically undistinguishable. Moreover, I have not found the chemical composition of Grt* in Table S2 and in Fig. 4c and Fig. 6. See also lines 368-370 and 444.

BULK-ROCK AND MINERAL COMPOSITION

  • In table 1 and in the text, the authors report the bulk-rock compositions of the three eclogite-types. In Fig. 3, they also plot bulk-rock composition of eclogite from Yangkou and unpublished data from Guanshan. However, in the discussion they do not try to explain the reasons of the chemical differences observable in Fig. 3. If they do not want to discuss about the protolith of the Sulu eclogites, it is sufficient that they report, in a supplementary table, only the bulk compositions used for the pseudosections.
  • Garnet composition. In my opinion, only Grt from sample XX18-5 and XX18-7 show relevant chemical zoning. Interestingly, but not described and interpreted, the zoning is asymmetric. I’m puzzled by Fig. 5. I do not understand what it should show and demonstrate.
  • Some EMP analyses are very bad. Total closure of hydrous minerals (and locally also of anhydrous ones) are too low. The A (i.e. X) site occupancy is locally very low. In some Grt, the Na content is too high. This raises doubts on the quality of the data and therefore on their meaningful use in figures or calculations…
  • A table reporting a selection of the most representative chemical analyses of minerals should be present in the main text (the tables showing the full data remain as supplementary materials)

PHASE EQUILIBRIUM MODELLING

  • In Fig. 9, 10, and 11 some metamorphic stages occur at P-T conditions well above the solidus of the system. However, the authors do not report petrographic evidence of partial (or even incipient) melting of the studied samples.
  • The thermodynamic modelling seems to indicated that the Grs content in garnet decreases with increasing pressure. I’m puzzeled, because usually the Grs content should increase at increasing pressure.

DISCUSSION and CONCLUSIONS

  • In Fig. 12, I do not understand why the proposed P-T path does not cross the stage R1
  • Actually, in my opinion, the authors do not clearly explain why the three eclogite-types record distinct P-T conditions (see above). Even Fig. 13 is not deeply discussed.
  • The authors write that the subduction was rapid and the exhumation was slow. However, they do not report geochronological data and neither clearly describe other evidences.

Other comments:

  • Line 45: “and differential thermobarometries were used”. What do you mean?
  • Line 49: “in detail”. So does this mean that some studies already occur? Which ones?
  • Line 74-75: “rootless with their long axes direction consistent with the general direction of regional gneissic schistosity”. I do not agree; actually, in Fig. 1b they are all oriented NNW-SSE and does not follow the antiform…
  • Lines 114-116: the data do not correspond to those shown in Table 2 (sample XX18-2 and 5)
  • Lines 131-132: some problems with the modal data…70+30+3+1+1 is higher than 100. The data do not correspond to those shown in Table 2 (sample XX18-7)
  • Lines 135-136: some problems with the modal data…55+35+10+0.2+1 is slightly higher than 100. This eclogite-type is not reported in Table 2.
  • Line 173: “unpublished data”: they should be reported in a supplementary table or removed by the figure 3.
  • Lines 192-194: I do not understand this sentence. Please, explain which are the differences and add a reference.
  • Line 195: “Eclogite-I is similar in composition to Guanshan eclogites…”. I do not agree. From Fig. 3b, it is evident that the Guanshan eclogites belong to the low-K tholeiitic series, whereas the Xiaoxinzhang eclogites make a trend belonging to the calc-alcaline series, although at the boundary to the low-K tholeiitic series.
  • Lines 214-220. General statements, they do not provide specific information.
  • Lines 226-227: what about omphacite in eclogite-III (see Fig. 6b)?
  • Line 302: “quini”. The right word should be “penta-variant”
  • Line 316. Why do you put 4 digits after the decimal point? With EMP analyses, the precision in the formula is at 3 digits.
  • Line 444 “Grt*”…I’m lost…I have understood that Grt* was stable during the prograde stages.
  • Lines 543 “Slow exhumation stage”. Line 555 “.. due to quick exhumation”. Contradiction.
  • 1a. The studied location is not reported. I suggest to increase the legibility of the figure removing the Dabie belt and enlarging the Sulu belt.
  • 1b. Please, increase the font size
  • 4b and d. I think that the Y scale is not correct (e.g., 0.05 instead of 0.5, etc.)
  • 7. (a) In the text, the authors do not distinguish between “Type I and II” amphiboles. (b) I do not understand the meaning of this diagram. I think that diagrams showing Si vs Mg/(Mg+Fe2+) or AlIV vs AlVI or AlIV vs (AlVI+Fe3+) could be more usefull.
  • 8. What do the blue line represent?
  • Supplementary tables: what do you mean by X (phase) and Y (phase)? In the legend present in table S1, more than a single line X and a single line Y should be present.

Reviewer 2 Report

The Sulu Orogenic Belt is a typical ultra-high pressure metamorphic terrane in the Tertiary, and is an ideal area in digging the process of continent-continent collision. In this manuscript, the authors used three representative eclogite samples of different types to elucidate the collisional process. Based on micropetrography, mineral chemistry and pseudosection modelling, they rebuilt the metamorphic P-T paths of these rocks and inferred that the Xiaoxinzhuang eclogites is different from those in other areas of the Sulu UHP terrane, and these rocks represent different rock slices during the subduction and exhumations. The evidences are solid and their inference is reasonable. Therefore, I recommend publication of this manuscript in Minerals subject to moderate revision.

Specific comments:

(a) Thermodynamic modeling strictly requires closed chemical behavior of any one sample. Therefore, the authors should specify that in the metamorphic process, the rock samples kept closed chemical system, especially in the prograde and retrograde segments.

(b) In sample XX18-2, chemical composition of the omphacite is nearly homogeneous, both for the inclusion preserved in the garnet and that occurs in the matrix (metamorphic peak). Similar phenomenon exists for the phengite. These are in contrast to that described in the abstract. Please check other minerals for other samples.

(c) We see the asymmetric chemical composition of the garnet in sample XX18-5 (Fig 4c). Please specify the reason.

Reviewer 3 Report

Dear Authors,

below you find some comments and suggestions for your paper “Ultrahigh-pressure metamorphism and P-T path of Xiaoxin-zhuang eclogites from the southern Sulu orogenic belt, eastern China, based on phase equilibria modelling” submitted to Minerals.

Line 15 – that = those

Line 16 – pseudosection = pseudosections

Line 17 – record = records

Line 21 – garnet enclosed by = garnet mantle of

Line 29 – different rock slices – various? Several? Or explain different from what.

Line 35 – benearth = beneath

Line 47 – eliminate “which”

Line 58 – THERMOCALC is not by Wei et al., please modify this citation. Maybe “by Powell et al, as used in Wei et al”, or “following Wei et al”, depending on what you mean.

Line 68 – eliminate “the”

Line 74 – please rephrase, I wonder how can be that a fault consists of gneiss and metasedimentary rocks. Do you mean the Sulu terrain? Which part of it?

Line 79 – what do you mean by the fact the mafic and ultramafic rocks are rootless? Maybe you can rephrase, it seems that these rocks make up boudins or bands. A description of the main structures involving the lenses and host rocks would add a lot to the paper, if possible. Field data are always extremely relevant to reconstruct PTt evolutions.

Line 83-86 – Are the ages of granites intruding gneiss known? Maybe it would be better to say that eclogite is within gneiss as lenses and/or bands, and that the gneiss have granitic to monzonitic protoliths? Se comment above.

Lines 98-100 – there are a bit too many minerals all together here, they are part of different parageneses.

Lines 106-120 – I suggest to describe textures also in the text, and not only in figure captions! Also on line 109 you indicate there is quartz formed from partial melt (as well as plagioclase on line 108 – by the way, if this is the analysis in table S3 there are problems, see later). This is way too little information, I urge you to put proper photographs of textures in order to demonstrate that there was melt, and describe them in detail in the text (something is in the discussion, move it here and write more than that. You should describe the evidences that make you think there was melt here.

Also you should make a figure only showing the texture of grt*, describing the texture in detail, and showing the differences in composition, using a compositional map, as this is a key to your findings, and this very small figures does not give the idea of the texture (are there minerals included in this garnet? Which ones?). Figure 2g doesn’t even have a scale (please add it). Please see also comments below at Figure 2.

Line 130 – here you state that amphibole is only retrograde, but then it is also present as prograde

General comment for paragraph 3

I suggest rewriting this paragraph, moving here some of the descriptions that are added later in the discussion sections, but that should be here (interpretations are OK in the discussion, but observations should be here). It would be helpful to the reader to have a table where the parageneses that are mentioned in the discussion are summarised (i.e. prograde, peak, retrograde 1 and retrograde 2), and that the textures showing these parageneses are clearly depicted in Figure 2. Furthermore Figure 2 can be divided in at least two figures, in order to show more clearly the important textures, such as that involving grt*, that at present is not clear enough to support your interpretation. It would enhance your paper a lot to show readable pictures of textures, and summarise the parageneses.

Line 129 – coesite occurs as inclusion in garnet: is that coesite? Or is it quartz, that you interpret having been former coesite? Please, be more specific in your description.

Lines 147-153 – please move this paragraph at the end of this section, to follow the logic of subsequent sections: the modelling is after whole rock and mineral composition.

Line 192 – eclogite-I samples have

Line 204 – eclogite-I samples are similar

Line 205 – is = are

Line 206 – is = are

Line 224 – decease = decrease

Line 234-235 – Omp-I shows higher j(o) (eliminate “much”)

Lines 253-254 - Phengite has Si = 3.15-3.42, (Mg + Fe2+) = 0.33–0.450 p.f.u., and Mg/(Mg + Fe2+) = 0.66–0.81

Line 257 – Other minerals = Epidote

Line 272-273 – are there carbonates? No mention in the microstructural description. Why do you need to account for carbonates? If there are some carbonates in the rocks, please say it in the description of samples.

Line 280-281 – this is a limiting choice, as demonstrated by Guiraud et al., 2001 (JMG). Maybe you can support this choice with some reasons, or at least advise readers that maybe water is not in excess and this would have bearings (and here you do not consider it).

Line 290 – add (TypeI) after the name of the sample in titles (here XX18-2).

General comment: I wonder why there are no photographs of typeII eclogite in the microstructural description (i.e. Fig. 2) please add some, and then there is no pseudosections of Type III eclogite but of TypeII eclogite.

Line 310 – Here there is a problem, as in the rock description you state that amphibole is retrograde (as demonstrated by textures) and do not mention the presence of amphibole as inclusions in garnets. Why would this assemblage represent a prograde stage for your rock, since there is not amphibole as inclusions? It doesn’t matter that the garnet composition stable in this field corresponds to that of your garnet if you don’t observe this assemblage. This is a major problem of your reconstruction! I’m referring to field “abcd” in figure 8, but the same applies to field “abcd” in figure 9a!

Line 328-331 – Here again there are no actual evidences in textures of the rocks of this reaction you are describing here. Can you make reference to textures? Otherwise say that you think this could be a possible way to abtain that result.

Lines 332-334 – this should go in the description of textures, where you must describe all evidences you observe in rocks. And where you need to add a table summarising the parageneses observed in each sample (not a list of all the observed minerals, as it is confusing and does not give informations about assemblages).

Add references about star-like quartz and wedge-shaped plagioclase as indicators of partial melting, then show these things in microphotographs in Fig. 2.

Line 336 and following – please see comment to line 310, this assemblage is not present in your observations, so you cannot use it, unless you discuss why should it be acceptable even though you do not observe amphibole in the rock paragenesis (garnet composition without the assemblage is not acceptable without a proper discussion).

Lines 370-372 – this is an affirmation that needs some more discussion….

Lines 376-383 – You need to explain how you reconstructed the bulk for this pseudosection, showing the calculations you undertake. This and the addition of a proper figure with the texture and compositional map of garnet and minerals around it would be important to sustain your affirmations with actual evidences (it is a pity not to put them here since you have them)

Line 387 – is the maximum Si content the one of phengite coexisting with the tabular garnet? This is essential. You should couple analyses of minerals that are in textural equilibrium.

Lines 392-395 – you cannot derive the PT condition from the composition of a single phase, you need to have a paragenesis. Please, provide the paragenesis, i.e. the minerals that are in equilibrium with this garnet.

Line 398 – for = with

Line 401 – Add a photo showing this assemblage (with evidences of melt presence) in Figure 2, where at present no Type II eclogite is represented.

Lines 431-434 – this introduction is obscure to me. I think there is a problem with English, could you please rephrase? Also minerals do not record per se different stages of metamorphism, but rather different assemblages can be preserved in the same rocks, and still we can apply an equilibrium view, provided we choose the equilibrium volumes wisely.

Line 435 – keep = be

Lines 442-445 – as already stated the data are not presented anywhere in the paper. Add a section “geothermobarometry” where you explain what you did, which data you used, how many points and some statistics – or at least a table with these data. Also discuss the textural position of the minerals you used for these calculations. Especially for the phengite discussed below (Lines 450-452). You should add a photo where phengite is in textural equilibrium with garnet g* to sustain your affirmations (see lines 453-455).

Lines 528-536 – are there any data about dating these rocks or the others you discuss?

Line 555-557 – there are no A, B and C in figure 12b, as instead suggested in the text, A, B and C are only in the other figure.

Line 558 – you should not call “exhumation rate” what apper to be geothermal gradients. In order to address exhumation rates you would need to have geochronological data (see above).

FIGURES and TABLES

Figure 2 – (Please see also comments in the text). I suggest

- to make a separate figure for Figure 2g, adding a scale, and possibly showing compositional maps for this garnet (or alternatively compositional profiles). If Figure 2g is an inset or part of Fig. 2e, please outline where it is placed.

- is Fig. 2e an enlargement of Fig. 2d? If so, please indicate where it is positioned in Fig. 2d

- there are no photographs of eclogite II textures, you should add some of these, as you later model some of these samples.

- there needs be some photographs of evidences of the presence of melt

Figure 4 – add to the left (Xalm, Xpy, Xgr) similarly to what you did on the right side of the figure

Table 4 – add a line to separate the calculated mineral compositions from the modal proportions of samples.

Table 5 – in this summary you mention geothermobarometers, but the data are not presented in the text. Please add a paragraph in which you mention what you did and the geothermobarometers you used, telling how many pairs of minerals, number of analyses and a bit of statistics you used. Also it would be good to have a supplementary table with some examples of analyses you used.

Tables S1, S2 and S3 – is grant = grain? Or what is the meaning of grant?

Table S3 - The An analysis has something wrong as it closes at 96%, which is rather bad for a plagioclase, SiO2 is too low, and the sum of cations is 5.12!!!! Please, put here a better analisys, or eliminate this one.

The same applies to the first garnet in Table S1, which closes at 7.95 and has very high SiO2, looking like a mixed analysis.

Please check the mineral analyses in tables, most of omphacite seems to close a bit low, and some of the garnets too. Any suggestions of why is this happening?

Round 2

Reviewer 1 Report

The manuscript has been extensively improved during revision, although the perspective of the work still remains local.

Below, I still report few comments.

INTRODUCTION

  • Lines 48-51 (see Point 2 of the reply): Ok. Please, add a reference from a review paper.

SAMPLING AND SAMPLE PETROGRAPHY

  • Line 102 (see Point 4 of the reply): The outcrop picture should be moved from Fig. 2k to Fig. 2a, accordingly to the position of the outcrop description in the main text.

BULK-ROCK AND MINERAL COMPOSITION

  • Line 204 (see Point 8 of the reply). Actually, the compositions of eclogite I split in two areas: i) in the area of the eclogites from CCSD (as the authors wrote); ii) in the area of the eclogites from Rongcheng (i.e., similar to the compositions of eclogite II and III).
  • Figure 4 (see Point 34 of the reply). Ah, ok…from a quick reading, I confused (Xsps) with the legend. So, I suggest to move the legend on top of Fig. 4a.

PHASE EQUILIBRIUM MODELLING

  • Figure 7 (see Point 36 of the reply). Ok. I suggest to specify the role of the blue line in the figure caption.

DISCUSSION and CONCLUSIONS

  • Line 333 and Figs 2a-c with caption (see Point 12). Please add a reference regarding these microstructures at line 333 and/or in the caption of Fig. 2. The melting microstructures in Fig. 2a and c are not so clear. Maybe pictures taken at cross polarized light could be helpful. Note that the partial melting of HP-UHP mafic lithologies is not a common process, so it has to be well documented. Did the authors observe microstructures similar to those reported at Sulu, General Hill locality (Wang et al. 2014, Nature Comm., 5, 5604; also quoted by the authors)?
  • Point 14. Ok. Please, make it clear in the text.
  • Point 15 and General comments. Ok. Please, make it clear in the text.

Reviewer 3 Report

Thanks for revising the manuscript.